



Measurement Report: Characterization of Aerosol Hygroscopicity over Southeast Asia during
the NASA CAMP[2]Ex Campaign
Genevieve Rose Lorenzo[1,2], Luke D. Ziemba[3], Avelino F. Arellano[1], Mary C. Barth[4], Ewan C.
Crosbie[3,5], Joshua P. DiGangi[3], Glenn S. Diskin[3], Richard Ferrare[3], Miguel Ricardo A. Hilario[1],
Michael A. Shook[3], Simone Tilmes[4], Jian Wang[6], Qian Xiao[6], Jun Zhang[4], and Armin
Sorooshian[1,7]
[1]Department of Hydrology and Atmospheric Sciences, University of Arizona, Tucson, Arizona,
85721, USA
[2]Manila Observatory, Quezon City, 1108, Philippines
[3]NASA Langley Research Center, Hampton, Virginia, 23681 USA
[4]Atmospheric Chemistry Observations & Modeling Laboratory, NSF National Center for
Atmospheric Research, Boulder, Colorado, 80301, USA
[5]Analytical Mechanics Associates, Inc, Hampton, Virginia, 23666 USA
[6]Department of Energy, Environmental & Chemical Engineering, Washington University in St
Louis, St. Louis, Missouri, 63130, USA
[7]Department of Chemical and Environmental Engineering, University of Arizona, Tucson,
Arizona, 85721, USA
*Correspondence to: armin@arizona.edu*
**Abstract**
This study characterizes the spatial and vertical nature of aerosol hygroscopicity in Southeast Asia
and relates it to aerosol composition and sources. Aerosol hygroscopicity via the light scattering
hygroscopic growth factor, f(RH), is calculated from the amplification of $PM_5$ aerosol ($D_p < 5$ μm)
scattering measurements from < 40% to 82% relative humidity during the Cloud, Aerosol, and
Monsoon Processes Philippines Experiment (CAMP[2]Ex) between August to October 2019 over
the northwest tropical Pacific. Median f(RH) is relatively low (1.26 with lower to upper quartiles
of 1.05 to 1.43) like polluted environments, due to the dominance of the mixture of organic carbon
and elemental carbon. The f(RH) is lowest due to smoke from the Maritime Continent (MC)
during its peak biomass burning season, coincident with high carbon monoxide concentrations (>
0.25 ppm) and pronounced levels of accumulation mode particles and organic mass fractions. The
highest f(RH) values are linked to coarser particles from the West Pacific and aged biomass
burning particles in the region farthest away from the MC, where f(RH) values are lower than
typical polluted marine environments. Convective transport and associated cloud processing in
these regions decrease and increase hygroscopicity aloft in cases with transported air masses
exhibiting increased organic and sulfate mass fractions, respectively. An evaluation of a global
chemical transport model (CAM-chem) for cases of vertical transport showed the
underrepresentation of organics resulting in overestimated modeled aerosol hygroscopicity. These
findings on aerosol hygroscopicity can help to improve aerosol representation in models and the
understanding of cloud formation.





## 1. Introduction

Aerosol particles affect climate and visibility through the direct and indirect extinction of solar radiation via absorption and scattering of light and through cloud formation, respectively. Aerosol hygroscopicity compounds aerosol effects on Earth's radiation budget (Zhao et al., 2018; Malm and Day, 2001), secondary aerosol formation and cloud formation (Köhler, 1936), and health (Dockery, 2001). Neglecting the effect of moisture on aerosol growth leads to incorrect estimation of the cooling at Earth's surface due to aerosol particles (Garland et al., 2007). For instance, a decrease in light extinction over the southeast U.S. was linked to reduced aerosol water uptake, coincident with decreases in the sulfate/organic ratio (Attwood et al., 2014). Particle aging/coating can cause underestimation of both aerosol hygroscopicity in the sub-saturated regime (Wang et al., 2018) and cloud condensation nuclei (CCN) activity in the supersaturated regime for aged particles in China by ~22% (Zhang et al., 2017). Remote sensing of aerosol optical properties is also affected by aerosol water content (Ferrare et al., 1998; Ferrare et al., 2023). Therefore, accurate aerosol hygroscopicity values are critical for remote sensing and satellite observation of aerosol particles (Ziemba et al., 2013; van Diedenhoven et al., 2022).

Aerosol hygroscopicity is described by physical quantities such as the diameter growth factor, g(RH), and light scattering hygroscopic growth factor, f(RH). The g(RH) parameter relates the wet particle diameter of the aerosol at a high relative humidity to the dry diameter of the aerosol at low relative humidity, while f(RH) relates total scattering due to the aerosol at high relative humidity (80%) to that at low relative humidity (<40%) (Waggoner et al., 1983; Hegg et al., 1993). Light scattering increases with relative humidity for most particles and is correlated to chemical composition and size (Baynard et al., 2006; Swietlicki et al., 2008) of particulate matter (Covert et al., 1972; Brock et al., 2016a).

Direct measurement of aerosol hygroscopicity, however, is difficult and is also not well-represented in climate models (Chen et al., 2014). The hygroscopicity parameter kappa, κ, is a single parameter that was developed to represent water uptake in models. It determines the volume (or mass or moles, with appropriate unit conversions) of water that is associated with a unit volume of a dry aerosol particle (Petters and Kreidenweis, 2007). A simple and commonly used water uptake model for calculating kappa is based on the Zdanovskii, Stokes and Robinson (ZSR) treatment for water soluble organic-inorganic mixed aerosol particles (Stokes and Robinson, 1966) where it is assumed that there are no interactions between the organic and inorganic species. In the ZSR model, water uptake of the individual non-interacting components can be summed up linearly to represent the total water uptake of the mixed aerosol. The interaction of organics and inorganics, however, along with the aging-specific density of organics is thought to influence hygroscopicity and affect both the ZSR calculation (Fan et al., 2020) and aerosol particle growth factor via changes in molecular structure, molecular weight, functionality, and/or other properties (Swietlicki et al., 2008).



Observed and simulated aerosol hygroscopicity using aforementioned parameters are greater in
clean marine air masses compared to air masses over land, near terrestrial biogenic sources which
are secondary organics precursors, and under polluted conditions (Swietlicki et al., 2008; Duplissy
et al., 2011; Petters and Kreidenweis, 2007).  In marine areas, hygroscopicity typically decreases
with altitude with decreasing inorganic fractions (Pringle et al., 2010). Cloud processing over
marine areas has been observed to increase the oxidation of organic aerosols (Che et al., 2022;
Dadashazar et al., 2022) and hygroscopicity in general (Crumeyrolle et al., 2008). Continental
aerosol particles have smaller diameters and are usually less hygroscopic due to more organic-rich
aerosol particles and pure elemental carbon (EC) particles (Wang et al., 2014; Kreidenweis and
Asa-Awuku, 2014). Organics are generally less hygroscopic than inorganics, and their
hygroscopicity is affected by oxidation level (e.g., O:C ratio), oxidation state, and solubility (Brock
et al., 2016a; Wu et al., 2016; Thalman et al., 2017). Aging has also been found to increase aerosol
hygroscopicity through the oxidation of secondary organic aerosols and organic aerosol
interactions with inorganics  (Engelhart et al., 2008; Liu et al., 2014; Saxena et al., 1995).
Although aerosol studies in the rapidly developing Southeast Asia (SEA) region  are increasing,
few are focused on the nature of aerosol particles and their interactions with water vapor and clouds
(Tsay et al., 2013; Ross et al., 2018; Reid et al., 2023). Understanding the interactions between
aerosols and the complex geographic, meteorological, and hydrological environment in Southeast
Asia remains challenging due to a still growing observational database, prevalence of clouds
interfering with remote sensing, and limited modeling studies (Tsay et al., 2013; Lee et al., 2018;
Chen et al., 2020; Hong and Di Girolamo, 2020; Amnuaylojaroen, 2023). This, along with
increased local and transported emissions and prevalent moisture-rich conditions in the region,
altogether motivate the need to understand aerosol hygroscopicity and associated impacts on
radiative transfer and on climate (Brock et al., 2016a; Ziemba et al., 2013). How freshly emitted
nearly hydrophobic particles transform into hygroscopic aerosol particles (Swietlicki et al., 2008),
for example, is an understudied topic in Southeast Asia where there are significant sources of
particles with low hygroscopicity (Reid et al., 2023). Understanding aerosol hygroscopicity will
also help in the need to improve remote sensing measurements in the region, which is affected by
overlapping high and low level clouds (Burgos et al., 2019; Hong and Di Girolamo, 2020).
Predicting aerosol hygroscopicity, especially at higher relative humidity (RH), is especially
difficult due to optical instruments underestimating particle light scattering at high RH and
mechanisms other than hygroscopicity impacting particle growth (Gasparini et al., 2006; Mochida
et al., 2006). This is important because atmospheric water content is high in Southeast Asia. For
example, the hygroscopicity of secondarily formed organics (via gas-to-particle conversion) is
found to be dependent on oxidation state for high RH (Shi et al., 2022). Southeast Asia has elevated
levels of organics, inorganics, and elemental carbon (Cruz et al., 2019; AzadiAghdam et al., 2019;
Oanh et al., 2006), allowing for an opportunity to see how hygroscopicity responds to a range of





relative values of each of these elevated components. Therefore, relating aerosol particle
composition to hygroscopicity, for closure (Xu et al., 2020), is particularly significant there.
Aerosol hygroscopicity is a crucial factor in the understanding and modeling of aerosol-cloud
interactions, because of the role hygroscopicity plays in cloud drop activation. The NASA Cloud,
Aerosol, and Monsoon Processes Philippines Experiment (CAMP$^2$Ex) was designed to understand
the role of aerosol particles in cloud formation and in regulating solar radiation during the
southwest monsoon (Reid et al., 2023). CAMP$^2$Ex occurred between August and October 2019
over the Philippines and neighboring areas. That campaign provides an aircraft dataset with
measurements focused on aerosol and cloud properties, which affords a valuable opportunity to
evaluate models with these measurements for shallow to moderate convection, which is one of the
biggest challenges for regional and global-scale atmosphere models to represent because these
clouds are much smaller than the model grid spacing.
In the CAMP$^2$Ex region, biomass burning aerosol hygroscopicity is over-estimated by global
atmosphere models simulating the CAMP2Ex campaign, with both the model size representation
of the aerosol particles and the size discrepancy between model and observations contributing to
this (Collow et al., 2022; Edwards et al., 2022). This overestimation can affect the representation
of clouds in the region. Clouds, especially shallow cumulus clouds, like those in the tropical West
Pacific (which the CAMP$^2$Ex region is part of) have been underestimated by models due to the
lack of observational data to improve convective parameterizations (Chandra et al., 2015). Part of
the challenge in modeling aerosol-cloud interactions is the ability to both have a high-resolution
representation, at the scale of the shallow convection, in a large enough domain, which is important
for understanding climatic effects to properly account for the bulk behavior of cloud fields (Spill
et al., 2019). This could be addressed by evolving modeling infrastructures, with higher resolution
schemes ranging from regional to convective scale within a larger domain (Pfister et al., 2020;
Radtke et al., 2021).
Knowledge gaps identified above are addressed in this study using the opportune CAMP$^2$Ex
dataset. To our knowledge, this is the first time this dataset has been explored extensively to
characterize aerosol hygroscopicity properties in the region. The goals of this proposed study are
to (i) characterize the spatial distribution of aerosol hygroscopicity in Southeast Asia during the
CAMP$^2$Ex airborne mission, (ii) relate aerosol hygroscopicity and composition, (iii) identify
emission events that impact aerosol hygroscopic growth, and (iv) evaluate a global chemical
transport model for aerosol vertical transport.
**2. Methods**
**2.1 CAMP$^2$Ex Field Campaign**
CAMP$^2$Ex included 19 research flights with a NASA P3 from 24 August to 5 October 2019.
Twelve of these flights were associated with the southwest monsoon (SWM) followed by a flow





reversal with seven flights conducted during the northeast monsoon (NEM) (Reid et al., 2023). As
summarized by Reid et al. (2023), the combination of airborne and ship-based measurements helps
to characterize interactions between various aerosol particle sources (e.g., biomass burning,
industrial, natural) and small to congestus convection. Below we note the P3 instruments most
relevant to this study.
**2.2 Observations and Derived Quantities**
**2.2.1 P3 Instrumentation**
As summarized in Table 1, data are used from a variety of aerosol particle and trace gas
instrumentation. The aerosol particle scattering and absorption instruments measure the optical
properties of bulk aerosol particles ($PM_5$): < 5 µm dry diameter (McNaughton et al., 2007). Aerosol
composition and size data for submicron particles were considered when they were collected using
a forward-facing shrouded isokinetic inlet. Non-refractory species in the submicron range studied
using an aerosol mass spectrometer (AMS) included sulfate, nitrate, ammonium, organics, and
chloride. We also use the ratio of the mass spectral marker m/z 44 marker relative to total organic
mass, $f_{44}$, as a possible indicator of air mass age. Submicron refractory species of black carbon
(also referred to as elemental carbon) from a single particle poot photometer (SP2) were also
included in the study.  Bulk water-soluble aerosol particles were collected using a particle-into-
liquid sampler (PILS) that was analyzed using ion chromatography; species quantified included
oxalate, $NH_4^+$, dimethylamine (DMA), $K^+$, $SO_4^{2-}$, $Ca^{2+}$, $Na^+$, $Mg^{2+}$, $Cl^-$, $NO_3^-$, $Br^-$, and $NO^-$. Cloud-
free conditions were identified to ensure the highest quality aerosol data using a cloud flag product
based on measurements from the fast cloud droplet probe (FCDP) and two-dimensional stereo
probe (2DS). Aerosol particle composition data were considered for those cases when the total
aerosol non-refractory particle mass was greater than 0.4 µg sm$^{-3}$.
**Table 1:** Summary of instrument data used in this work.



| Parameter | Instrument | Time Resolution | Uncertainty | Sampled Aerosol Particle Size | Reference |
|---|---|---|---|---|---|
| Latitude, Longitude, and Altitude | Northrop Grumman Litton 251 EGI | 1 s | ~5 m spherical error probable; 0.01° | N/A | Reid et al., 2023 |
| Dry (RH < 40%) and humidified (RH = 80%)) light scattering coefficient ($\lambda$ = 450, 550, 700 nm) | Parallel humidified TSI 3563 Nephelometers | 1 s | 30% | < 5μm diameter | Ziemba et al., 2013; McNaughton et al., 2007 |
| Dry (RH < 40%) light absorption coefficient ($\lambda$ = 470, 532, 660 nm) | Radiance Research 3 Particle Soot Absorption Photometer (PSAP) | 1 s | 15% | < 5μm diameter | Mason et al., 2018; McNaughton et al., 2007 |
| Non-refractory aerosol (Organics, $SO_4$, $NO_3$, $NH_4$, Cl) mass concentration | Aerodyne High-Resolution Time-of-Flight Mass Spectrometer (AMS) | 25 s | LLOD ($\mu g\ sm^{-3}$): Organics, 0.169; $SO_4$, 0.039; $NO_3$, 0.035; $NH_4$, 0.169; Cl, 0.036; Uncertainty: 50% | approximate relevant size range is 60-600 nm vac. aero. diameter | DeCarlo et al., 2008 |
| Water-soluble mass concentration | Particle-into-liquid sampler (PILS) followed by offline ion chromatography analysis | 238 s | 30% | < 5μm diameter | Sorooshian et al., 2006 Crosbie et al., 2020 |
| Refractory black carbon (BC) mass concentration | Single Particle Soot Photometer (DMT SP2) | 1 s | 10% | 100 - 700 nm diameter | Schwarz et al., 2006 |
| CO concentration | Picarro G2401-m | 1 s | 5 ppb | N/A | DiGangi et al., 2021 |
| Volume and number concentration of particles | TSI Laser Aerosol Spectrometer (LAS) Model 3340 | 1 s | 20% | 0.1 - 5 μm optical diameter | Moore et al., 2021 |
| Volume size distribution | Fast integrated mobility spectrometer (FIMS) | 1 s | Concentration: 15%; Size: 3% | 10 – 500 nm mobility diameter | Kulkarni and Wang, 2006; Wang et al., 2017; 2018 |



| Cloud particles | SPEC-Hawkeye FCDP | 1 s | 50% | $2 - 50$ μm diameter | Knollenberg 1981, Lawson et al. 2017; Woods et al. 2018 |
| Cloud particles | SPEC-Hawkeye 2DS | 1 s | 20% | $10$ μm – $3$ mm diameter | Lawson et al. 2006a, Woods et al. 2018 |


### 2.2.2 Aerosol Hygroscopicity, f(RH), and Other Aerosol Optical Properties

Aerosol hygroscopicity is reported using the parameter f(RH), which is unitless and is the
amplification factor in scattering due to a change in RH. The f(RH) parameter is calculated from
the empirically derived exponential fit coefficient, gamma ($\gamma$), at 20% ($RH_{dry}$) and 80% ($RH_{wet}$)
relative humidity (Ziemba et al., 2013). The gamma parameter is based on measurements of
scattering at 550 nm at two different relative humidity levels: dry ($< 40\%$) and humidified
(controlled to $82 \pm 10\%$). Gamma (Eq. 1) was calculated for times where the dry ($SC_{dry}$) and
humidified ($SC_{wet}$) scattering coefficients (Table 1) are greater than or equal to 5 Mm$^{-1}$. The f(RH)
(Eq. 2) was then derived from $\gamma$ (Hänel, 1976).

$$\gamma \;= \frac{\ln \frac{SC_{wet}}{SC_{dry}}}{\ln \frac{100 - RH_{dry}}{100 - RH_{wet}}} \qquad (1)$$


$$f(RH) = (\frac{100 - RH_{wet}}{100 - RH_{dry}})^{-\gamma} \qquad (2)$$


All nephelometer scattering coefficient measurements were corrected for truncation errors using
the Anderson and Ogren's method (Anderson and Ogren, 1998). Relative humidity measurements
for the calculation of f(RH) were calibrated in the laboratory using nebulized ammonium sulfate
deliquescence at 80% (Brooks et al., 2002). System response is verified in flight by introducing
hydrophobic polystyrene latex spheres into the sample stream to ensure an f(RH) value of 1.0 is
observed. Absorption measurements were corrected for a variety of errors using the method from
Virkkula (2010) (Virkkula, 2010). Note that sampling efficiency decreases for supermicron
diameter particles with increasing size up to the 5-µm inlet cutoff, due to losses in transport tubing
and in the drying/humidification control system. Thus, derived f(RH) is applicable to
accumulation-mode particles and is partially sensitive to coarse-mode particles from 1-5 µm
diameter.

Single scattering albedo (SSA) was calculated when both the scattering and absorption coefficients
(smoothed with 30 s running average) were greater than 2 Mm$^{-1}$. The Ångström exponent (AE)
was calculated using the smoothed 30 s running average of the scattering and absorption
coefficients. The scattering Ångström exponent (SAE, 450-700 nm) was computed when the
scattering coefficient was greater than 2 Mm$^{-1}$ based on Ziemba et al. (2013) and the absorption





Ångström  exponent (AAE, 470-660 nm) was computed when the absorption coefficient was
greater than 2 Mm$^{-1}$ (Mason et al., 2018).
**2.2.3 Sea Salt**
Bulk sea salt mass concentration was calculated using summed PILS concentrations of Na$^+$, Cl$^-$,
and Mg$^{2+}$ along with the respective concentrations of the mass concentrations of K$^+$, Ca$^{2+}$ and
SO$_4^{2-}$ in sea salt (0.037, 0.04 and 0.25, respectively, by mass) (Crosbie et al., 2022).
**2.2.4 Aerosol Particle Classification**
Aerosol particle optical data were grouped depending on their AAE and SAE (Section 2.2.2)
values and based on the method of Cazorla et al. (2013), which used sun photometer measurements
to arrive at the following classifications as part of evaluating aircraft data over California, USA:
coated large particles, dust and elemental carbon mix, dust dominated, organic carbon and dust
mix, organic carbon dominated, elemental carbon and organic carbon mix, mixed, and elemental
carbon dominated. This method has been used for cases when chemical composition is not
available (Höpner et al., 2019), and can be useful because the composition data from the AMS is
limited to submicron particles (0.06 to 0.6 µm vacuum aerodynamic diameter) while the f(RH)
includes relatively larger particles (< 5 µm dry diameter).
**2.2.5 Effective Particle Density**
Following Shingler et al. (Shingler et al., 2016b), effective particle density was calculated by
dividing the sum of the 30 s total mass concentration from the AMS species (organics, SO$_4^{2-}$, NO$_3^-$
, NH$_4^+$, and Cl$^-$) and 30 s averaged black carbon from SP2 by the 30 s averaged integrated (for
particles with diameter from 0.1 to 1 µm) volume concentration from the LAS (Table 1).
**2.3 Modeling**
**2.3.1 Trajectory Analysis**
This work leverages trajectory results explained in detail by Hilario et al. (2021). This data product
associates air masses undergoing long-range transport nearby from the Maritime Continent (MC),
East Asia (EA), peninsular Southeast Asia (PSEA), and the West Pacific (WP) with specific
CAMP$^2$Ex flight locations, where air masses were within the regions at altitudes below 2 km for
more than 6 h. The source regions are approximately within the following lowest and highest
latitude and longitude values, respectively: MC (-9.5º - 6.5º and 95º - 119º), EA (22º - 47º and 105º
- 121.5º), PSEA (10º - 20º and 98º - 106 º), and WP (3º-25º and 120.5º-122.5º). The National
Oceanic and Atmospheric Administration Hybrid Single Particle Lagrangian Integrated Trajectory
Model (HYSPLIT) (Stein et al., 2015; Rolph et al., 2017) was used to produce 5-day back
trajectories. The classification "Other" was used for back trajectories that either passed by the
regions but at elevations above the boundary layer (defined as 2 km), came from sources farther
away than the four listed above, were more localized to the Philippines, had too few sample counts,
or were from stagnant air (Hilario et al., 2021).



**2.3.2 CAM-chem Model Configuration**

The Community Atmosphere Model with comprehensive tropospheric and stratospheric chemistry
CAM-chem is used here as the atmospheric component of the Community Earth System Model
(CESM2). CAM-chem includes the modal aerosol model (MAM4) (Liu et al., 2016) and results
were evaluated with the CAMP²Ex AMS composition and f(RH) observations for two case studies.
Two CAM-chem simulations with different horizontal resolutions have been performed using the
spectral element grid mesh and dynamical core. The grid mesh resolution for one simulation was
uniform with ~111 km (labeled ne30). The other simulation employed regional refinement over
East Asia with grid spacing ~27 km (labeled MUSICA) in the regionally refined region and ~111
km elsewhere across the globe; a configuration as part of the Multi-Scale Infrastructure for
Chemistry and Aerosols version 0 (MUSICAv0) (Schwantes et al., 2022). More information about
CAM-chem is in the supplementary section (S1).

In CAM-chem, aerosol hygroscopicity is represented with the kappa value (Petters and
Kreidenweis, 2007) using the mixing rule (Stokes and Robinson, 1966). Kappa is calculated from
the CAM-chem Aitken (0.015 – 0.053 µm) and accumulation mode (0.058 – 0.48 µm) outputs
based on the volume fractions of the aerosol constituents (ε, Eq. 3) that was derived from their
densities (Table S1). The following internally mixed aerosol species from CAM-chem were
included in the analysis: organics (primary/hydrophobic, aged/hygroscopic, and secondary
($C_{15}H_{38}O_2$)), sulfate ($NH_4HSO_4$), sea salt, dust ($AlSiO_5$), and black carbon (primary/hydrophobic
and aged/hygroscopic) (Tilmes et al., 2023). Some limitations in the calculation of kappa are that
CAM-chem does not include nitrate aerosol, and that constant kappa values for primary and aged
organics based on past work (Table S1) were used, even if it is known that kappa of organics varies
with aging (Kuang et al., 2020). Kappa was also calculated from the submicron AMS and SP2
observations using the assigned species properties for available aerosol species: aged organics,
aged black carbon, and ammonium sulfate. Submicron sea salt and dust data are not available and
were not included in the calculation for submicron kappa from observations.

$$\kappa_{chem} = \sum \kappa_i \, \varepsilon_i \qquad\qquad (3)$$

To enable a corresponding evaluation based on actual aerosol hygroscopicity observations, the
f(RH) from CAMP²Ex were converted to kappa based on past studies (Brock et al., 2016b; Kuang
et al., 2017; van Diedenhoven et al., 2022). The wet (80%) and dry (20%) relative humidity values
of the scattering measurements were used to convert f(RH) to an optical kappa ($\kappa_{opt}$) based on
Equation 4 (Brock et al., 2016b; Kuang et al., 2017). This was approximated from the proportional
relationship between the aerosol scattering cross section (which is the basis of f(RH)) and aerosol
volume (the change of which is usually described as growth factor) which Brock et al. (2016b) in
their study had associated with kappa. The optical kappa was converted to the chemical kappa
($\kappa_{chem}$) based on the slope of the relationship between $\kappa_{opt}$ and $\kappa_{chem}$ from Brock et al. (2016b). This
method (Eq. 5) of converting $\kappa_{opt}$ to $\kappa_{chem}$ was also used by van Diedenhoven et al. (2022) for the
f(RH) data for CAMP²Ex and is associated with 40% uncertainty (van Diedenhoven et al., 2022).





The derived chemical kappa values for bulk aerosol particles are compared to the kappa calculated
using the ZSR mixing rule from the submicron observations (AMS and SP2) and model outputs.

$$f(RH) = \frac{1+\kappa_{opt}\frac{RH_{wet}}{100-RH_{wet}}}{1+\kappa_{opt}\frac{RH_{dry}}{100-RH_{dry}}} \quad (4)$$

$$\kappa_{chem} \approx \frac{\kappa_{opt}}{0.56} \quad (5)$$

**3. Results and Discussion**
**3.1 General Characterization of f(RH) for CAMP²Ex**
**3.1.1   Spatial and Vertical Distribution of f(RH)**
The f(RH) values in the CAMP²Ex campaign (Fig. 1a) were relatively low (median of 1.24 for
143,107 1 s points) with a narrow range of values (25th percentile (Q1) of 1.05 and 75th percentile
(Q3) of 1.42). The distribution of f(RH) and ambient RH values is narrowest in the lowest altitudes
(< 3 km), where most of the samples were taken (Fig. 1b). Both f(RH) and RH distributions become
broader at higher altitudes, where there is a higher prevalence also for higher f(RH) values and
lower RH values. Broadening of the f(RH) distribution at higher altitudes is at least partially
attributed to lower dry scattering coefficients that result in increased uncertainty in the calculation
of f(RH). The different aerosol sources (Fig. 1a) in the region also contribute to the range of
measured f(RH), with transported emissions from more distant sources presumably more
influential at the highest altitude ranges.

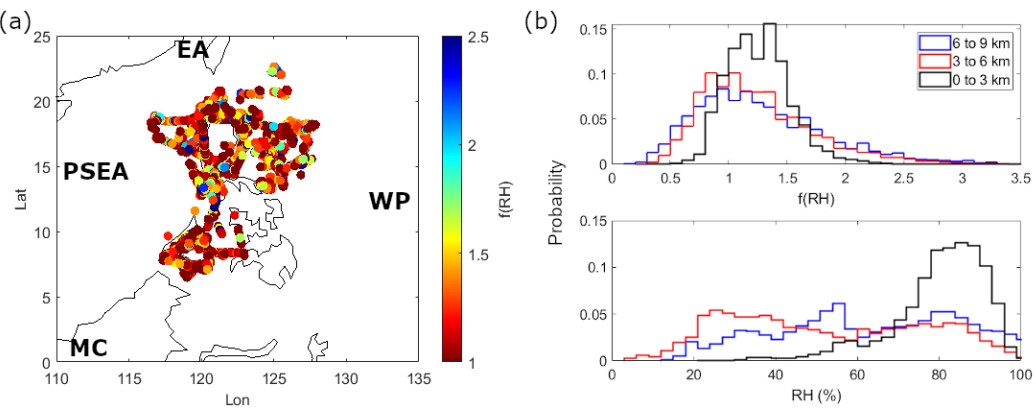


Figure 1. (a) Map showing f(RH) 1 s values along the CAMP²Ex flight paths with approximate locations and air mass
sources: Maritime Continent (MC), East Asia (EA), peninsular Southeast Asia (PSEA), and the West Pacific (WP)
(PSEA is farther west at ~105°E and EA extends farther north) of air mass sources (Hilario et al., 2021) around the
region. (b) Histograms of f(RH) and relative humidity (RH) for three altitude bins with the following counts: n$_{(0\ to\ 3}$
$_{km)}$: 139,026, n$_{(3\ to\ 6\ km)}$: 10,321, and n$_{(6\ to\ 9\ km)}$: 7,260.
The lowest median f(RH) (1.05 with Q1 and Q3 of 0.94 and 1.2) is from air masses traced to the
MC (Fig. 2a), which coincide with influence from smoke particles. The air mass from EA (Fig.



2a) has the narrowest range of values (Q1: 1.28, median (Q2): 1.38, Q3: 1.47), likely representative
of urban aerosol particles. The highest median f(RH) (1.49 with Q1 and Q3 of 1.26 of 1.73) are
from air masses with back trajectories from the WP (Fig. 2a), likely due to marine aerosol particles
interacting with particles from the MC and other regional sources. This mixing of the otherwise
clean marine air with regional pollution sources effectively decreases aerosol hygroscopicity and
this type of environment is often called a polluted marine environment (Titos et al., 2021).
Most (98%) of the f(RH) data were calculated for observations below 3 km (Fig. 2b), due to the
relatively clean free troposphere in the region. Median f(RH) values generally decrease with
altitude in the lower 3 km. An increase in median f(RH) is observed between 4 – 6 km (median
from 1.14 to 1.32), where the contribution of the mixed air mass ("Other") is most dominant. The
f(RH) values decrease above 6 km to the lowest median f(RH) (1.07) between 8 – 9 km, where air
masses are generally from Other and the MC.
Latitudinally (Fig. 2c), f(RH) is lowest nearest Borneo in the MC (median of 0.95 to 1.07 in the
regions from 6.85° N – 10.25° N), coincident with the dominant influence of biomass burning.
There were active fires in the area during the time samples were taken from this latitude. f(RH) is
highest in the northern Philippines (median of 1.33 to 1.44 in the regions from 18.75° N – 22.15°
N), where the influence of WP and EA air was most prevalent. This is consistent with the longitude
data (Fig. 2d), which exhibit the lowest f(RH) values for longitudes (median of 1.02 to 1.18 in the
regions from 117.95° E – 120.55° E) that had the highest MC contribution. The highest f(RH) is
observed in longitudes (median of 1.38 to 1.40 in the regions west of 116.65° E and east of 123.15°
E) that were more associated with the northern Philippines (Fig. 1a). To delve deeper into our
analysis, we discuss next the size and composition data.



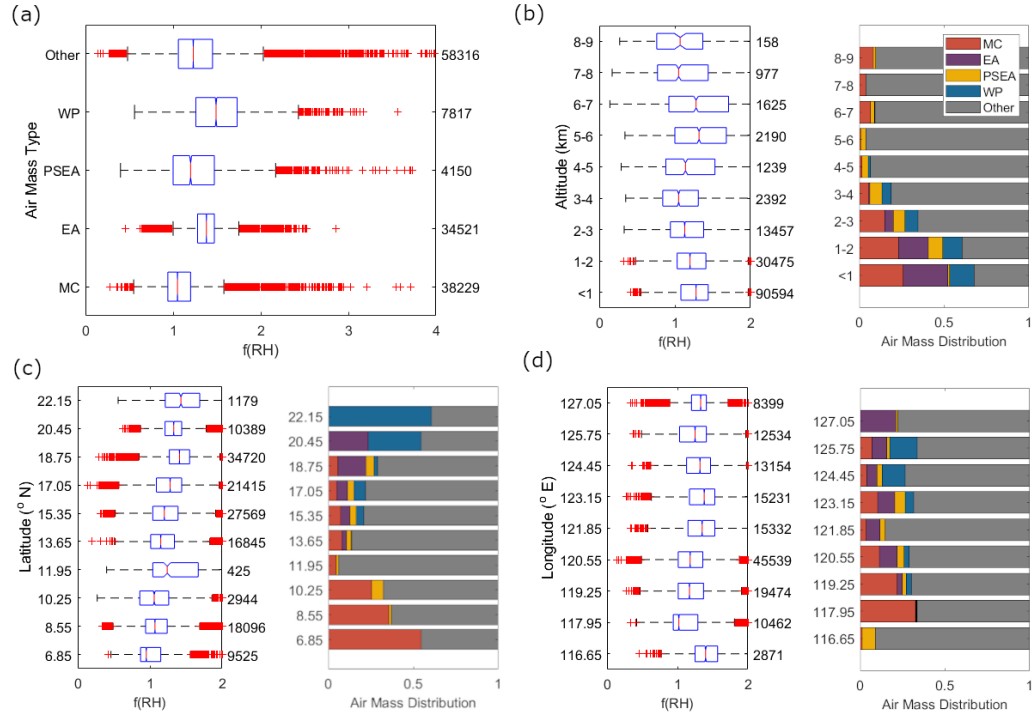

Figure 2. Distribution of 1 s f(RH) data during the CAMP²Ex field campaign for relevant (a) air masses, (b) altitude, (c) latitude, and (d) longitude levels and corresponding stacked bars of air mass contributions for panels b - d. The (a) air mass types are from Hilario et al. (2021). The numbers to the right of the notched boxplots are the counts, and the (b – d) bars to the right of the boxplots show the fractional contribution of each air mass type to the total number of air masses in the specific location. The y-values for (c-d) latitude and longitude are the midpoints of the specific coordinate bins.

### 3.1.2   f(RH) Relationships with Size and Composition

The relative size of particles per air mass can be inferred from the extinction Ångström exponent (AE), which relates the extinction of light at specific wavelengths to particle size (Ångström, 1929), where larger AE suggests smaller particles. Since scattering is the dominant component of extinction, the scattering Ångström exponent (SAE) is often used to relate to particle size. The median SAE values (Fig. 3a) are similar and between 2.09 (MC with Q1 and Q3 of 1.92 and 2.26) to 2.16 (Other with Q1 and Q3 of 1.75 and 2.32) for four of five air masses, indicative of smaller particles. The WP has a median value of 1.37 (with Q1 and Q3 of 1.05 and 1.88), suggestive of the presence of a mixture of accumulation-mode and coarse-mode particles (SAE < 1 occurs for large particles like sea salt and dust) (Schuster et al., 2006; Bergstrom et al., 2007). Past studies have suggested that biomass burning particles exhibit SAE values greater than 1.4, which does not discount the fact that most of the f(RH) data collected during CAMP²Ex is possibly impacted by biomass burning.



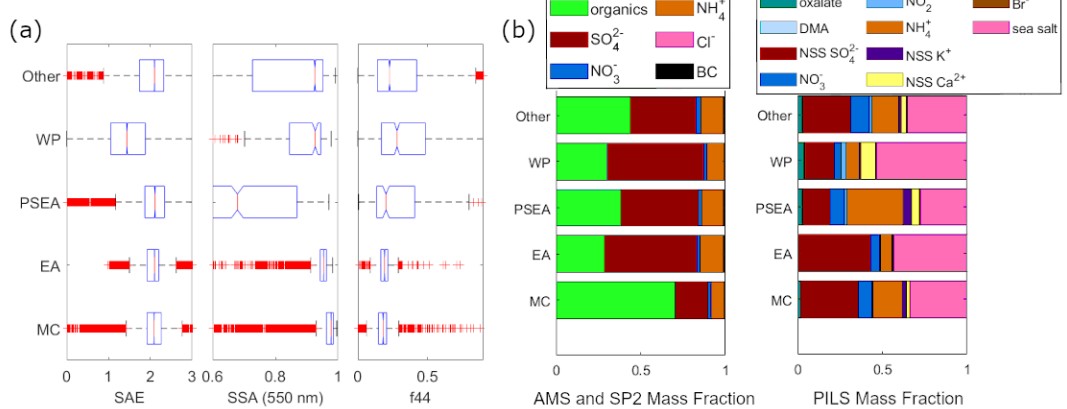

Figure 3. (a) Boxplots of 1 s scattering Ångström exponent (SAE), single scattering albedo (SSA, 550 nm), and ratio
of the mass spectral marker m/z 44 marker relative to total organic mass ($f_{44}$) and (b) (left) submicron mean organic,
$SO_4^{2-}$, $NO_3^-$, $NH_4^+$, $Cl^-$, and black carbon (BC) mass fractions and (right) bulk (< 5 µm) oxalate, $NH_4^+$, dimethylamine
(DMA), non-sea salt $K^+$, non-sea salt $SO_4^{2-}$, non-sea salt $Ca^{2+}$, $NO_3^-$, $Br^-$, $NO_2^-$, and sea salt mass fractions per air mass
(Hilario et al., 2021).
One of the indicators of biomass burning in Southeast Asia is organic matter (Adam et al., 2021).
The MC air mass (Fig. 3b) has the greatest (71%) mass fraction of organics among the air masses
and the highest median SSA (0.98 with Q1 and Q3 of 0.96 and 0.99) (Fig. 3a) that suggests more
scattering, rather than absorbing, particles (Moosmüller and Sorensen, 2018). This is consistent
with observations from field work in tropical peatland fire in Southeast Asia, where particles were
mostly from smoldering combustion and were moderately absorbing, with brown carbon
dominating absorbance (Stockwell et al., 2016). Smoldering combustion, which is more common
in the Maritime Continent (Reid et al., 2013), is known to produce less elemental carbon (Reid et
al., 2005) and potassium (Robinson et al., 2004) compared to flaming combustion. Biomass
burning activities were active in the MC during the field campaign (Reid et al., 2023). The MC air
mass also has the lowest $f_{44}$ median values (0.18 with Q1 and Q3 of 0.15 and 0.21) (Fig. 3a)
suggesting it is the least-oxidized and less photochemically-aged air mass and thus closest to the
source compared to other air masses. Chen et al. (2022) who studied tropical peat smoldering
(similar to those in MC) showed that primary organics were not oxidized ($f_{44}$ < 0.02) while
secondary organics were highly oxidized, that oxidation increases $f_{44}$ (oxidized organics: 0.01 <
$f_{44}$ < 0.11), and that high RH speeds up the oxidation process especially for smaller particles (~100
nm) (Chen et al., 2022). In the U.S., as another example, $f_{44}$ from wildfire plumes up to 8 hours
old did not exceed 0.12 (Garofalo et al., 2019). The high RH during CAMP[2]Ex (Fig. 1b) likely led
to the increased oxidation of secondary organics resulting in median $f_{44}$ values that are relatively
high (for all air masses even in MC), suggesting that most of the particles sampled during
CAMP[2]Ex are aged. The high organic mass fraction is consistent with the MC having the lowest
median f(RH) as organics are known to reduce aerosol hygroscopicity (Kalberer, 2014; Sorooshian
et al., 2017; Shingler et al., 2016a).
There is an inverse relationship between hygroscopicity and organic mass fraction data that is most
evident in f(RH) values collected within the boundary layer (< 3 km) during CAMP[2]Ex (Fig. 4a).



Those data points that have the highest organic mass fraction values (> 0.6) are also associated
with CO concentrations (> 250 ppb) that are typically associated with biomass burning (Shingler
et al., 2016a) and are mostly from the MC area (Fig. 4b). The slope of the inverse relationship
between f(RH) and organic mass fraction is most steeply negative (-0.81) for the air masses coming
from EA (Fig. 4c) .

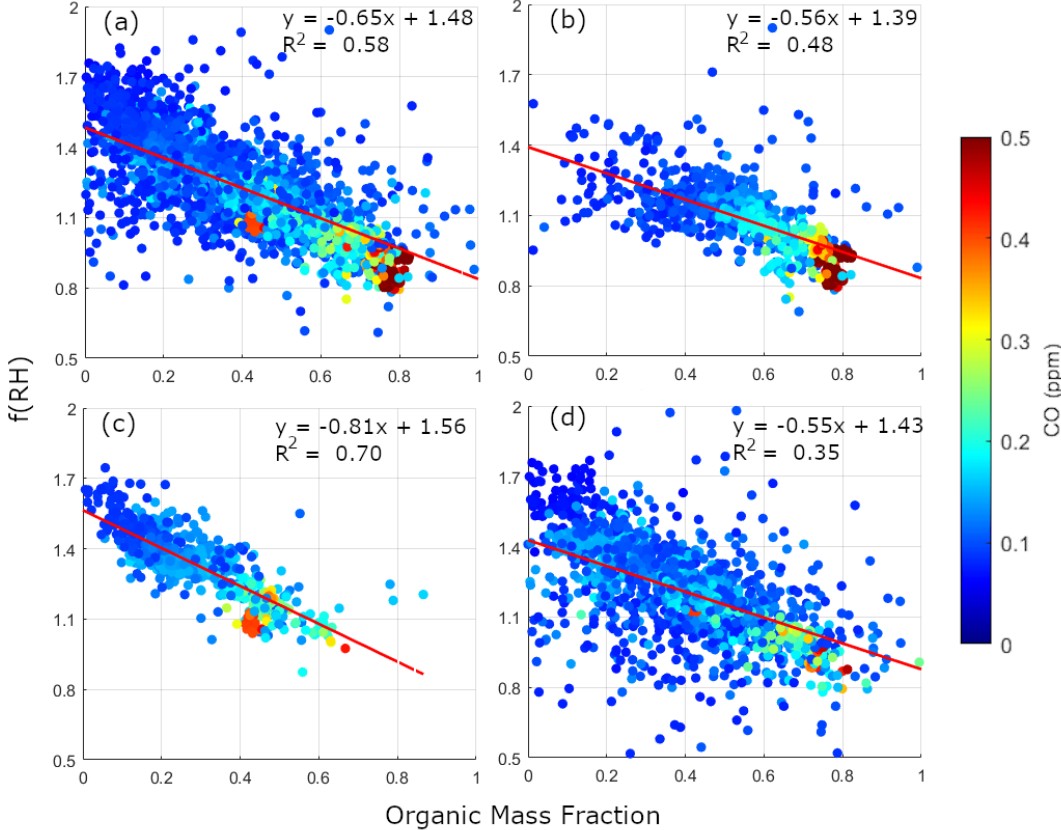


Figure 4. (a-d) Scatter plots of 30 s averaged f(RH) and corresponding organic mass fraction contribution to total
submicron aerosol mass (sum of organic, $SO_4^{2-}$, $NO_3^-$, $NH_4^+$, $Cl^-$, and black carbon (BC) mass concentration) for (a)
all data collected at altitudes below 3 km and (b-d) for air masses coming from (b) the Maritime Continent (MC), (c)
East Asia, and (d) Other. The dots are colored by their CO concentration.
The highest median f(RH) values are from the WP and EA air masses (Fig. 2a). Both have lower
organic mass fractions (0.29 and 0.30, respectively), but have distinctly larger aerosol size profiles
based on their SAE values (Fig. 3a). The presence of coarser particles from the WP, based on its
marine origin (with relatively higher DMA, Fig. 3b) and high sea salt fraction (54%, Fig. 3b),
contributed to it having the highest median f(RH) amongst the air masses. The WP air masses
appear to have interacted with aged organic particles from biomass burning and industry including
fine absorbing particles potentially from industry and dust owing to its relatively high median $f_{44}$





(0.26 with Q1 and Q3 of 0.17 and 0.48) and low median SSA (0.93 with Q1 and Q3 of 0.85 and
0.95). Particles with predominantly clean marine sources tend to have higher (≥ 0.97) SSA
(Dubovik et al., 2002), thus the WP data suggest a polluted marine source (Lacagnina et al., 2015).
Though, we note that the air masses from the WP have relatively low scattering and absorption
coefficients and that these could have affected the calculations for SSA. EA air exhibits the highest
non-sea salt sulfate mass fraction (0.43) (Fig. 3b) that, along with its predominantly fine particle
size and median SSA (0.95 with Q1 and Q3 of 0.94 and 0.96), strongly suggests that it was
transported from an urban source. Sulfate is a known industrial product of East Asia (Smith et al.,
2011; Li et al., 2017; Lorenzo et al., 2023).
Particles from PSEA have the lowest median SSA (0.68 with Q1 and Q3 of 0.55 and 0.87) (Fig.
3a). This suggests the presence of more absorbing particles relative to scattering, possibly
including elemental carbon and aged dust. Based on the PSEA air masses having the highest non-
sea salt potassium mass fraction (0.05, Fig. 3b) and highest dust-EC mix among air masses, the
particles could be from biomass burning. It is well-documented that dust can be entrained in smoke
plumes due to reasons such as turbulent mixing around flames and burn fronts (Palmer, 1981;
Kavouras et al., 2012; Maudlin et al., 2015; Schlosser et al., 2017). Dust from East Asia and
biomass burning from PSEA have been observed to be mixed in the boreal spring in Taiwan (Dong
et al., 2018), and though CAMP²Ex sampled during a different season, it is still possible that this
mixing of East Asian dust and PSEA biomass burning could have occurred and impacted the
CAMP²Ex region during the field campaign.

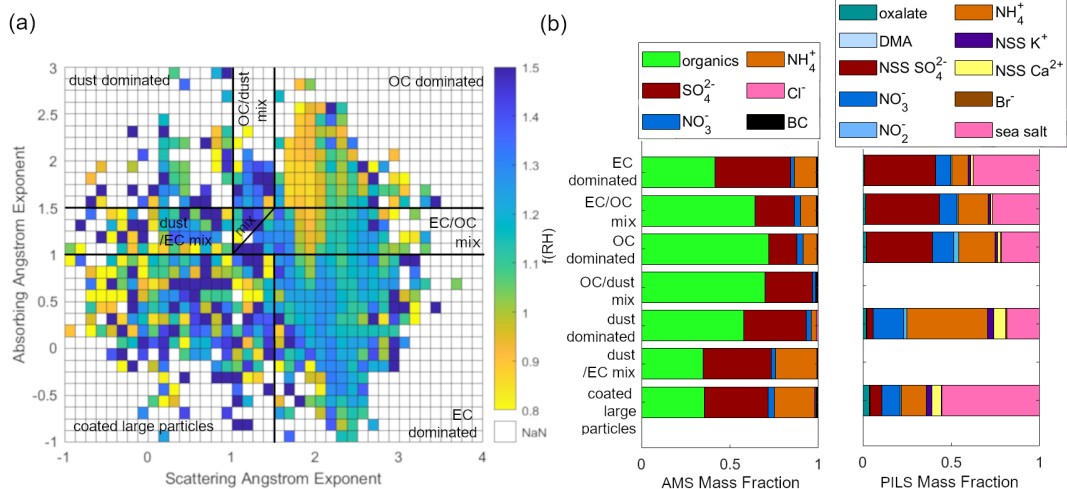

Figure 5. (a) Median f(RH) of data points with absorbing Ångström Exponent (AAE) and scattering Ångström
Exponent (SAE) values that correspond to suggested aerosol types from past studies (Cazorla et al., 2013) with
following counts and bulk median f(RH) per aerosol type: EC dominated (16,908 and 1.19), EC/OC mix + mix (20,686
and 1.03), OC dominated (1942 and 0.94), OC/dust mix (55 and 1.31), dust dominated (79 and 1.15), dust/EC mix
(204 and 1.12), and coated large particles (729 and 1.21). (b) (left) Submicron mean organic, SO₄²⁻, NO₃⁻, NH₄⁺, Cl⁻,
and black carbon (BC) mass fractions and (right) available bulk (< 5 µm) oxalate, NH₄⁺, dimethylamine (DMA), non-
sea salt K⁺, non-sea salt SO₄²⁻, non-sea salt Ca²⁺, NO₃⁻, Br⁻, NO₂⁻, and sea salt mass fractions per aerosol type.



Using the AAE and SAE with categorization determined by Cazorla et al. (2013), the f(RH) can
be related to the types of aerosols. The highest median f(RH) of 1.31 (with Q1 and Q3 of 0.89 and
1.46) and 1.21 (with Q1 and Q3 of 0.88 and 1.68) are from the OC/dust mix and coated large
particles aerosol types (Fig. 5), respectively, though there are only 55 data points for the OC/dust
mix so the aerosol classification for it may not be as robust. As such, there are no PILS
compositional data available for the OC/dust mix aerosol type. However, from available data from
dust dominated particles, it is also possible that hygroscopic particles like ammonium and nitrate,
which have the greatest bulk mass fractions (0.45 and 0.17) in the dust dominated particles
compared to other aerosol types, partitioned to dust and increased f(RH) for the OC/dust mix
particles. The high f(RH) for the coated large particles is consistent with its the largest bulk sea
salt mass fraction (0.56) (Fig. 5b) compared to other aerosol types. Note that the median f(RH)
values for the aerosol types may be slightly different than the raw 1 s f(RH) data because the
median f(RH) is only calculated for the aerosol types when both the scattering and absorbing
Ångström Exponent values are available.
Most of the aerosol particles (< 5 µm dry diameter and 93% of all particles) collected during the
CAMP²Ex field campaign have optical properties (Fig. 5a) that resembled EC/OC mix + mix (in
this case we combined the sub-categories of EC/OC mix and mix) (51%) and EC dominated (42%)
aerosol types. The EC dominated particles have the third to the highest median f(RH) at 1.19 (with
Q1 and Q3 of 1.05 and 1.33), which is unusual because EC is known to be hydrophobic.
Compositional data sheds some light on this, because the particles classified as EC dominated have
relatively higher submicron sulfate (0.43) and bulk sea salt mass (0.38) fractions (Fig. 5b)
compared to particles classified as OC (0.16 and 0.22) and dust dominated (0.35 and 0.19), which
are known to have low hygroscopicity in general. This mixing of sulfate and sea salt with EC
dominated particles increases their bulk f(RH). The EC dominated particles come from the EA,
MC, and Other air masses. East Asia is a known sulfur source and shipping contributes to sulfate
in the region (Miller et al., 2023). Peat smoke particles from MC have also been found in past
studies to have sulfate mixed with carbonaceous species (Nakajima et al., 1999). The presence of
OC (0.64 submicron mass fraction) decreased the median f(RH) values (Fig. 5b) where the median
f(RH) for the EC/OC mix + mix type was 1.03 (with Q1 and Q3 of 0.94 and 1.18).
The OC dominated type have the highest submicron organic mass fraction (0.72) and a median
f(RH) of 0.94 (with Q1 and Q3 of 0.85 and 1.06), which is consistent with the expected effect of
organics to lower hygroscopicity. The median f(RH) values below 1, though, are counterintuitive
given that the presence of moisture in the region is thought to generally increase the particle size
and consequently the amount of aerosol light scattering. Data where f(RH) is less than unity have
been observed in past studies due a number of suspected factors including (but not limited to)
volatilization effects, changes in optical properties during humidification, and    particle
restructuring (Shingler et al., 2016b).



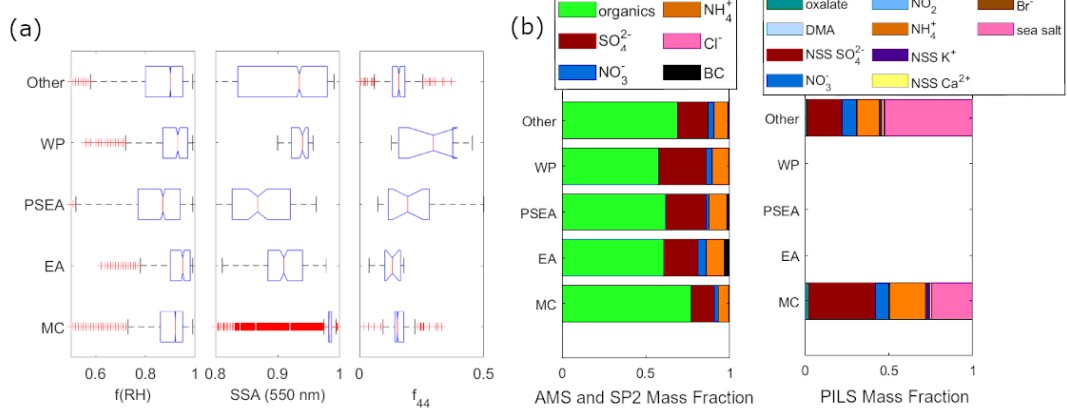


Figure 6. (a) Boxplots of 1 s f(RH) < 1 (MC: 14,612, EA: 473, PSEA: 986, WP: 483, and Other: 9919 counts) and the corresponding single scattering albedo (SSA, 550 nm) and ratio of the mass spectral marker m/z 44 marker relative to total organic mass ($f_{44}$) and (b) (left) submicron mean organic, $SO_4^{2-}$, $NO_3^-$, $NH_4^+$, $Cl^-$, and black carbon (BC) mass fractions and (right) available bulk (< 5 µm) oxalate, $NH_4^+$, DMA, non-sea salt $K^+$, non-sea salt $SO_4^{2-}$, non-sea salt $Ca^{2+}$, $NO_3^-$, $Br^-$, $NO_2^-$, and sea salt mass fractions per regional air mass (Hilario et al., 2021) for times with f(RH) < 1.

To investigate aerosol characteristics when f(RH) is < 1, we plot the f(RH) box plots from each of the regions for data points where f(RH) < 1 (Fig. 6a). There are several instances (26,473 times or 19% of the time for the whole campaign) when the f(RH) was below 1. In general, organic mass fractions greater than 0.75 correspond to sub-1 f(RH) (Fig. 4a). The most prevalent regional air mass association for sub-1 f(RH) is from the MC (f(RH) < 1 from the MC was 56% of all data). The organic mass fraction is dominant (0.57 – 0.77), and almost doubled compared to the whole campaign (Fig. 3b), for all the air masses with sub-1 f(RH) (Fig. 6b) and is highest for the MC (0.77). The organics from the sub-1 f(RH) data are the least aged throughout the campaign for the MC, PSEA, and Other air masses (Fig. 6a). The CAMP²Ex data offer an opportunity to inspect the prevalence of such values and to see what factors coincide with such situations. However, the unique sample make-up of the particles in the CAMP²Ex region makes other reasons, including sample losses due to volatilization, also plausible (Reid et al., 2023). Shingler et al. (Shingler et al., 2016b) also observed such sub-1 values for both f(RH) and the humidified diameter growth factor g(RH) in air masses enriched with carbonaceous components over North America.

The most dominant aerosol types for the sub-1 f(RH) data are the carbonaceous ones (96%) (Fig. 5b), with the EC/OC component contributing the most: EC/OC mix + mix (the combined sub-categories of EC/OC mix and mix) (68%), EC-dominated (18%), and OC-dominated (10%). Both bulk ammonium and sulfate mass fractions for the MC also are higher for the sub-1 f(RH) cases (Fig. 6b) compared to the whole campaign (Fig. 3b, PILS), while the bulk sea salt mass fraction was lower. The bulk sea salt mass fraction of the Other category (Fig. 6b) increases for the sub-1 f(RH). Most of the particles contributing to the sub-1 f(RH) are fine particles (median Ångström Exponent ~2, Fig. 6a) with more reflective characteristics (SSA ≥ 0.90) compared to data for the



entire campaign. To understand more about sub-1 f(RH), we will look at a selected case in the
succeeding section.

**3.2    Case Studies**

**3.2.1 Sub-1 f(RH) from Biomass Burning Smoke**

The chosen case study is part of a flight on 16 September 2019 that occurred closest to the Maritime
Continent, which is the source of the air masses that had the most counts of sub-1 f(RH) (Section
3.1.2). This flight coincided with an active biomass burning event on 14 September 2019 (NASA,
2020) and fire hotspots in the Maritime Continent were numerous throughout September 2019
(Othman et al., 2022). The back trajectories at 01:00 UTC, when the aircraft began to make
measurements closest to the surface (~300 m altitude) and perpendicular to the wind, all come
from the southwest of the aircraft location (Fig. 7a), in the direction from where biomass burning
emissions were being transported. Details about the flight and conditions during this case study
are found in Crosbie et al. (2022). In their study they note the smoke to be from Kalimantan with
an age between 48 and 72 h. Consistently low f(RH) (all below 1, Fig. 7b) values were observed
for a little less than an hour, until the aircraft began its ascent (Fig.7b). The aerosol mass
concentrations of >70 µg m$^{-3}$ were among the highest in the entire field campaign and were
dominated by organics (Fig. 7b).

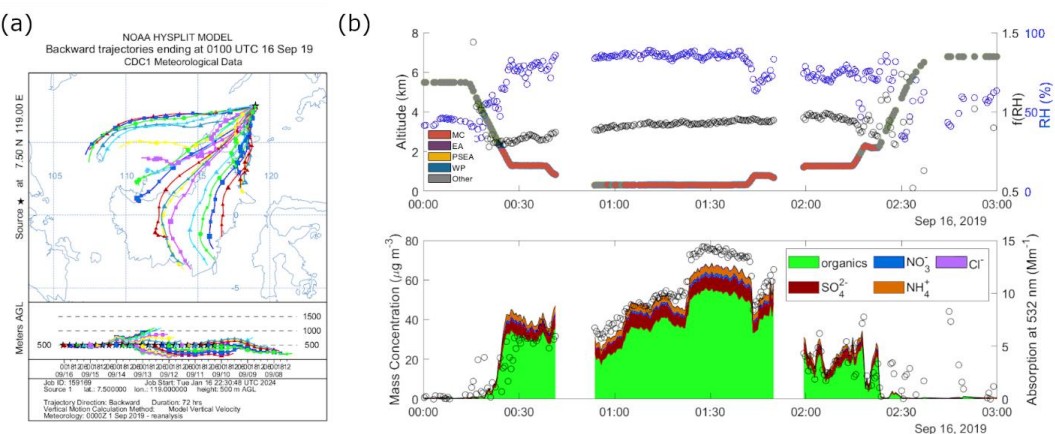


Figure 7. Case study of sub-1 f(RH) on 16 September 2019. (a) Five-day back trajectories at the approximate flight
location at 01:00 UTC. (b) (top) Time series plots of altitude colored by regional air mass, f(RH) in black circles, and
RH (blue circles), and (bottom) aerosol mass concentrations from the AMS and absorption at 532 nm (black circles)
during one of the flight legs closest to the surface from 00:00 to 03:00 UTC.
The sub-1 f(RH) during the lowest altitudes of the aircraft, from 01:00 to 01:40 UTC, are correlated
with approximations of the submicron particle effective density (Fig. 8a). Absorption
measurements along with back trajectories linked to active fires point to the increasing presence
of EC and brown carbon with increasing particle density (even if there were no valid EC





measurements for this specific time). Submicron number concentration, dry absorption (532 nm),
organic mass fraction, and single scattering albedo are highest for the smallest effective density
values for sub-1 f(RH) (Fig. 8b) during the entire campaign, likely due to the presence of EC and
OC. This is consistent with the most dominant aerosol type during the field campaign, which is a
mix of elemental and organic carbon based on SAE and AAE values. The dataset cannot be used
to prove particle restructuring and very likely the sub-1 f(RH) values are due to other factors since
particles' history included passes through humid areas where restructuring presumably would have
already occurred prior to reaching the aircraft. In addition, smoldering from peat, which is
dominant in the MC fires (Reid et al., 2023), is known to produce homogenous spheres with no
voids and similar to OC (Pokhrel et al., 2021), making restructuring unlikely to be the dominant
mechanism to explain our findings. However, we present these results for the sake of documenting
a Southeast Asia case of these unusual events occurring in EC/organic-rich air masses similar to
past work (e.g., Shingler et al., 2016b and references therein) to emphasize that these events occur
throughout the world and to motivate more research into the matter.

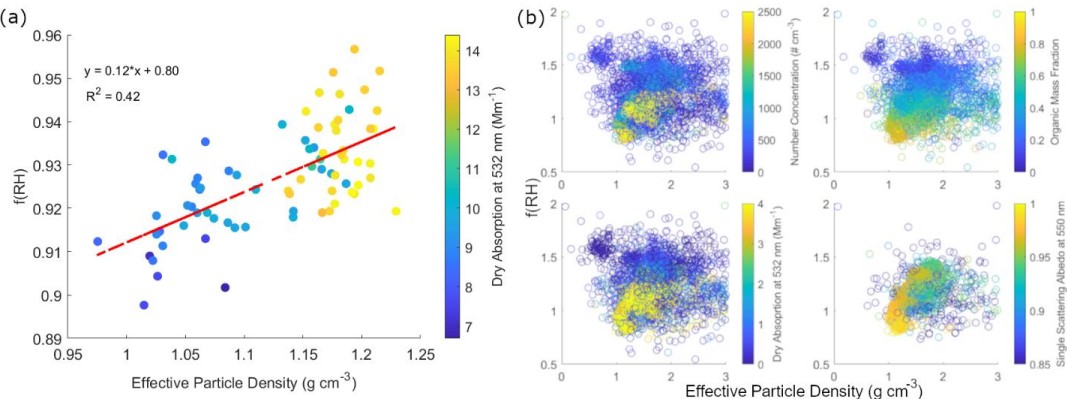


Figure 8. 30 s f(RH) and effective particle density plots for (a) 16 September 2019 from 01:00 to 01:40 UTC colored
by dry absorption at 532 nm, and (b) for the entire CAMP$^2$Ex field campaign and colored by (clockwise from top left)
LAS number concentration of particles with diameters between 100 and 1000 nm, organic mass fraction, single
scattering albedo at 550 nm, and dry absorption at 532 nm.
Smoke particles are known to have a range of density values, depending on their degree of
atmospheric aging, affecting their size, and the processes they undergo. The effective particle
density is usually lower for biomass burning particles from smoldering that have larger diameters
(Pokhrel et al., 2021), compared to flaming. Freshly emitted smoldering particles have effective
densities from 1.03 g cm$^{-3}$ to 1.21 g cm$^{-3}$ that do not vary much with diameter based on a laboratory
study (Pokhrel et al., 2021). Our calculated values fall within this range (Fig. 8). This adds
confidence to our observation that organics (brown carbon), which dominate smoldering
emissions, are the major contributors to the sub-1 f(RH). Both aerosol hygroscopicity and effective
particle density are important for properly modeling cloud condensation nuclei, one of the most
important factors in aerosol-cloud interactions.





**3.2.2 Vertical Transport**

The vertical distribution of aerosols affects cloud formation, we investigate this through two cases of aerosol vertical transport. The vertical transport cases were identified from averages of the available vertical profiles made during CAMP²Ex. A large-scale event (01:53 to 06:20 UTC 20 Sep 2019) north of Luzon, Philippines, due to a tropical cyclone, and a smaller-scale event (02:55 to 06:02 UTC 24 Sep 2019) east of Luzon, due to shallow convection were chosen for the case studies. The measured median CO mixing ratio was used as a tracer for vertical transport (Kar et al., 2004). An increase in the median CO mixing ratio at higher altitudes along with multi-level winds from a similar direction were the main criteria used to identify the cases.

Northerly to northwesterly winds are the dominant source for the first case (Fig. 9a) influenced by tropical cyclone Tapah (TC Tapah's center was ~600 km northeast of the aircraft location). This suggests the influence of East Asia on the sample as described in Crosbie et al. (2022), where the meteorology is discussed in more detail. There is a general decrease in f(RH) from the lower levels that follows a similar trend to the decrease in total mass (Fig. 9b) and sulfate mass fraction. Sulfate is hygroscopic so it is understandable that the f(RH) decreased with altitude as the sulfate decreased. The increase in CO between 6-7 km (Fig. 9b) suggests vertical transport aloft and was accompanied by a subsequent decrease in f(RH) and an increase in organic mass fraction (Fig. 9b) between 6-8 km. This is consistent with analysis in the previous sections, which show decreased f(RH) with increased organic mass fractions.

Accompanying the general decrease of f(RH) with altitude are decreasing submicron median volume size distribution (VSD) magnitudes and volume weighted average diameters (Fig. 10). Possible reasons for these trends are that larger hygroscopic particles, such as sulfate (which make up the largest mass fraction at lower levels), were scavenged (especially at lower altitudes) and/or activated into cloud drops, leaving the smaller particles behind. Submicron aerosol mass also decreases with altitude, with values below 1 µg m$^{-3}$ (Fig. 9b). There is a slight increase in VSD from 6-8 km compared to 4-6 km, and especially at larger diameters, broadening the VSD curve which may suggest cloud processing (Eck et al., 2012). Though sulfate enhancements have traditionally been the marker for cloud processes (Barth et al., 2000; Faloona, 2009), more recent studies have observed potential cloud processing cases with increased organics (i.e. Wonaschuetz et al., 2012; Dadashazar et al., 2022). This case could thus be showing vertically transported organic matter that has possibly affected or was affected by the clouds between 6-7 km. Another observation is the increase of ammonium and nitrate mass fractions above 4 km (Fig. 9b). There is no corresponding valid f(RH) data, and we can only infer based on the CO profile that begins to shift toward a more positive slope with increasing altitude that this increase in ammonium could also be associated with cloud processing. Ammonia is abundant in East Asia (Pawar et al., 2021) And although there are still limited studies, the scavenging efficiencies of organics and ammonium (compared to sulfate) could also be contributing to their increasing mass fractions with altitude (Yang et al., 2015; Hilario et al., 2023).

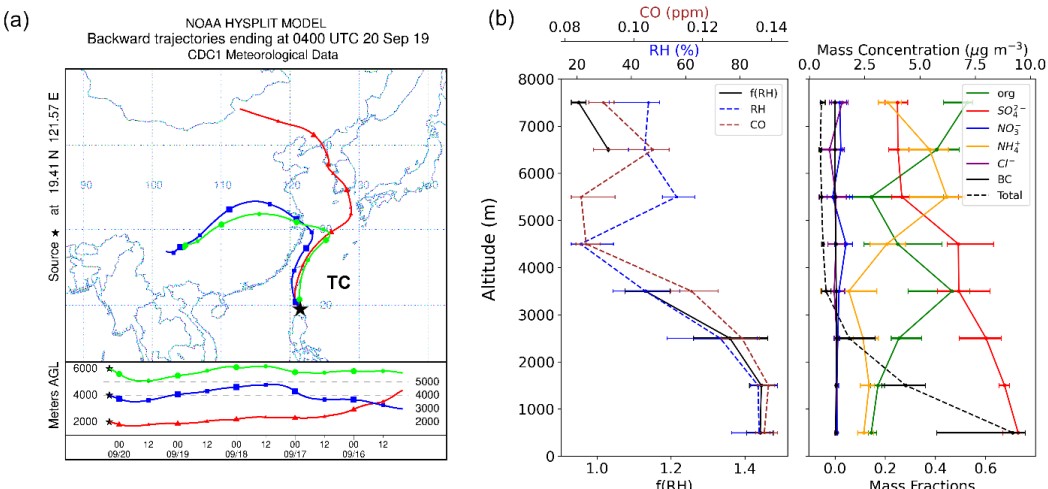

592

Figure 9. Case study of tropical cyclone-induced convection on 20 September 2019. (a) Five-day multi-level back
trajectories from the average location of the aircraft from 01:53 to 06:20 UTC with the approximate TC center location
and (b) vertical profiles (median, 25 and 75[th] percentiles) of (left) f(RH), relative humidity (RH), and CO mixing ratio
and (right) submicron mass fractions of organics, sulfate, nitrate, ammonium, and black carbon and sum of mass
concentrations from AMS and SP2 (dashed black line) for 1 km altitude bins.

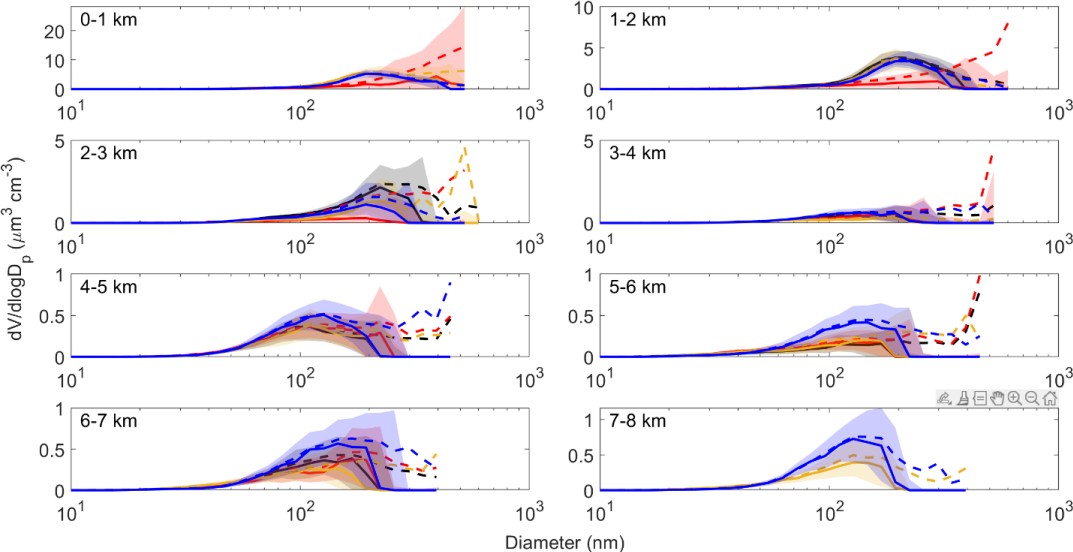

598

Figure 10. Median (solid line), mean (dashed line), and 25[th] to 75[th] percentiles (shaded area) of volume size
distributions of submicron particles (FIMS) every 1 km altitude for the case study of tropical cyclone-induced
convection on 20 September 2019 from 01:53 to 06:20 UTC. The colors represent four separate vertical profiles,
where black was from 01:35 to 02:19 UTC, red was from 03:40 to 04:44 UTC, yellow was from 04:59 to 05:27 UTC,
and blue was from 05:52 to 06:20 UTC.



The second case, the shallow cumulus case on 24 September 2019 (Hilario et al., 2023),  has multi-level wind trajectories initially from the West Pacific in the northeast direction, which appear to have come from the Philippines and the general southwest direction from two days before (Fig. 11a). The most evident increase in CO (Fig. 11b) is observed between 5-5.5 km, at an altitude lower than the previous case. Both RH and f(RH) have a similar trend to CO throughout the vertical profile, which slightly follows the trend of sulfate mass fraction. This is observed especially in the lower levels (until ~2 km) where RH and f(RH) are relatively steady, even with a large decrease in total aerosol mass.

At higher altitudes, above 4.5 km, f(RH) increases to a greater degree with decreased organic mass fraction and increased sulfate mass fraction (Fig. 11b), possibly due to cloud processing.  Like the first case, the VSD plots (Fig. 12) for this case show decreasing VSD magnitudes and volume weighted average diameters with increasing altitudes. The broadening of the VSDs above 3 km, concurrent with the increased decreased sulfate mass fraction, likewise suggests cloud processing. There is a known sulfate source, a power plant in western Luzon, in the Philippines that is along the path of the l.5 km back trajectory (Fig. 11a) and may possibly contribute to sulfate in the region (Lorenzo et al., 2023). It is possible that this and other sulfate sources in the region, such as those from ships (Miller et al., 2023), are being transported vertically and affecting aerosol hygroscopicity in the areas where there is shallow convection.

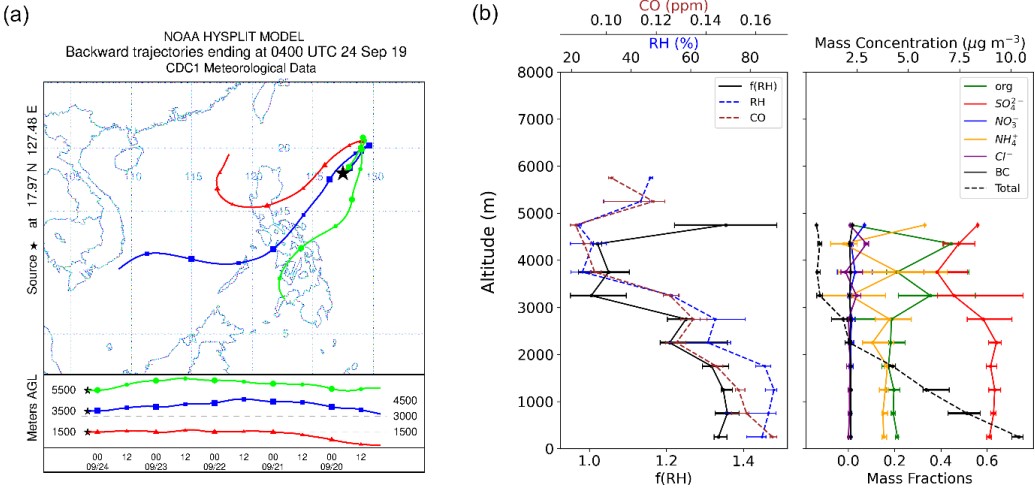

Figure 11. Case study of shallow convection on 24 September 2019. (a) Five-day multi-level back trajectories from the average location of the aircraft from 02:55 to 06:02 UTC and (b) vertical profiles (median, 25 and 75[th] percentiles) of (left) f(RH), relative humidity (RH), and CO mixing ratio and (right) submicron mass fractions of organics, sulfate, nitrate, ammonium, and black carbon and sum of mass concentrations from AMS and SP2 for 0.5 km altitude bins.





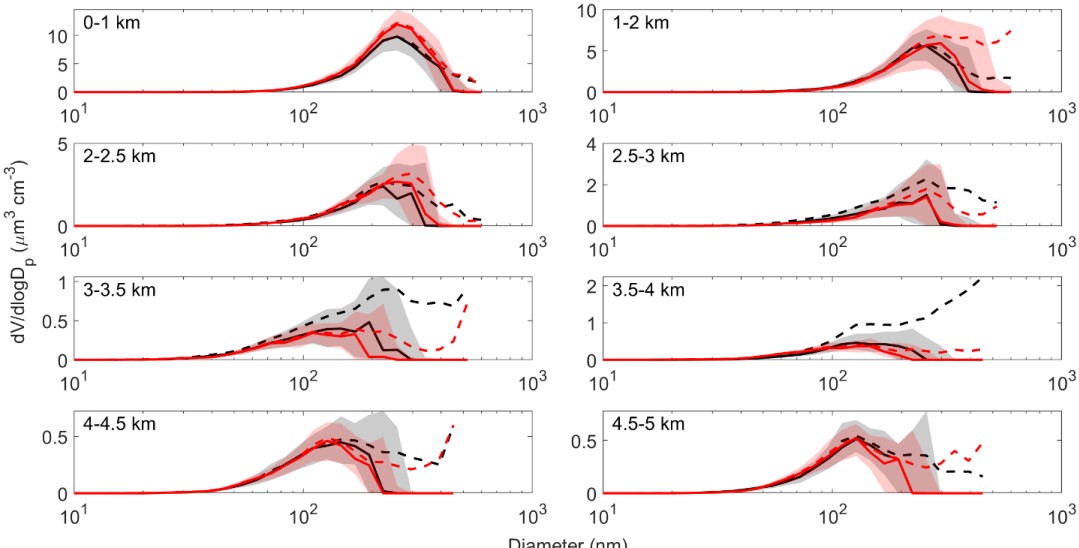

Figure 12. Median (solid line), mean (dashed line), and 25[th] to 75[th] percentiles (shaded area) of volume size distributions of submicron particles (FIMS) in 0.5-1 km altitude increments for the case study of shallow convection on 24 September 2019 from 02:55 to 06:02 UTC. The colors represent two separate vertical profiles, where black was from 02:55 to 04:07 UTC and red was from 05:42 to 06:02 UTC

In summary, for both cases vertical transport in cumulus clouds results in higher f(RH), lower sulfate mass fraction, and higher organic mass fraction at cloud outflow altitudes. The VSDs and the averaged diameter of the aerosol size decrease from cloud base to cloud outflow altitudes, likely due to cloud processing. The understanding and representation of the vertical transport or aerosols due to the tropical cyclone and shallow convection and their role in aerosol-cloud interaction is further investigated as we evaluate modeled data using the two cases that we have just discussed.

### 3.2.3  Model Evaluation

To give regional context to the cases, we begin the discussion of model evaluation with the horizontal distribution of the modeled (~111 km resolution) near-surface aerosol hygroscopicity (kappa: calculated from the volume fractions and kappa values (Eq. 3) of the modeled submicron aerosol species as described in 2.3.2)  and winds within the CAMP[2]Ex domain. We will compare the model kappa values to kappa from different regions around the world based on previous studies. Subsequently, we will discuss how the modeled kappa at different vertical levels compare to the kappa derived from bulk f(RH) (< 5 µm) (Eqs. 4 and 5) as well as from kappa derived from submicron AMS and SP2 species (Eq. 3) from the aircraft observations.

The CAM-chem modeled surface kappa values (Fig. 13) in the CAMP[2]Ex region of interest, during the time when the three cases discussed in 3.2.1 and 3.2.2 were sampled, are within the range of globally modeled surface kappa in marine (0.9 – 1.0), continental (0.1 – 0.4), and continental





outflow (0.4 – 0.6) regions from previous studies (Pringle et al., 2010). The highest modeled kappa
(~1.0, Figs. 13a and 13b) is like those previously found in marine areas and is influenced by the
high winds, coming from the direction of the Pacific Ocean, and can increase sea salt emissions.
This is due to the tropical cyclone which was still developing on 16 September 2019 and more
fully developed on 20 September 2019. The Pacific Ocean also is the source of high kappa (~0.9
- 1.0) for the shallow convection case on 24 September 2019 (Fig. 13c). For all three cases there
was low kappa (< 0.4) in areas with low wind speeds and over land especially over Borneo and
East Asia, typical of continental and continental outflow regions. For the two cases that were
within the southwest monsoon (Figs. 13a and 13b) the lowest kappa values (< 0.2) were in Borneo
and downwind areas including Southern Philippines and areas east of it.
The modeled surface kappa (~0.1 – 0.5) below the aircraft positions for the case studies of interest
(red lines in Fig. 13), though over the sea, are lower than in those areas most affected by the Pacific
Ocean and are influenced by emissions from Borneo (Figs. 7a and 13a), East Asia (Figs. 9a and
13b) and the Philippines (Fig. 11a and 13c) based on back trajectories. Kappa, calculated from the
median f(RH) derived from scattering measurements, for the five air masses discussed in sections
3.1 and 3.2 (MC: -0.02, EA: 0.14, PSEA: 0.10, WP: 0.24, and Other: 0.11), all fall in the
continental category even though the majority of these air masses are technically in regions with
continental outflow. The CAM-chem kappa values for the maritime continent are very close to
zero, but in the aircraft positions for the case studies with vertical transport (Figs. 13b and 13c) the
CAM-chem kappa values (~0.40 – 0.50) are more than double the range of calculated kappa from
East Asia (0.14) and Other (0.11) air masses that are influencing the case studies. To make sense
of this difference between globally modeled surface kappa values and that which is calculated from
the CAMP²Ex f(RH), we delve more into aerosol vertical transport and its connection to aerosol
hygroscopicity and evaluate the vertical profile of the kappa calculated from the CAM-chem model
outputs.

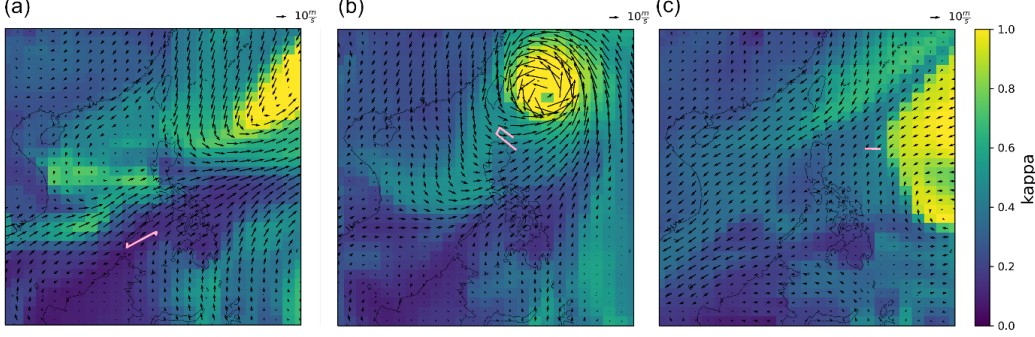


Fig. 13. Modeled kappa and winds at 957 hPa for (a) the biomass burning smoke case at 06:00 UTC on 16 September
2019, (b) the tropical cyclone-induced case at 06:00 UTC on 20 September 2019 and, (c) the shallow convection case
at 06:00 UTC on 24 September 2019. The pink lines represent the aircraft locations from 02:15 to 04:22 UTC on 16
September 2019, 01:53 to 06:20 UTC on 20 September 2019 and, from 02:55 to 06:02 UTC on 24 September 2019.



The CAM-chem model (Fig. 14a) was able to represent the general trends in the observed total
submicron mass vertical profile for the tropical cyclone-induced convection case. The observations
from AMS are only for non-refractory aerosol species, however, and so sea salt and dust are not
included in the total mass, possibly affecting the comparison with the model output, which
accounts for both. The uniform ~111 km grid mesh CAM-chem and MUSICA outputs show
approximate contributions by sea salt and dust to be 5 to 20% of the total mass (Figs. 15b and 15c).
This could explain the relatively higher total mass values from the model outputs compared to
observations for some altitude levels, though in general even without dust and sea salt the CAM-
chem model still has slightly higher aerosol total mass values (dashed lines in Fig. 14a) compared
to the observations. The CAM-chem model represents the approximate aerosol total mass well for
this case probably because of its large-scale nature, where wind speeds were relatively high and
where the aerosol particles are potentially from a large source (East Asia) (Figs. 9a and 13a).
Together, the sum of the sulfate and ammonium mass fractions are similar for the observed (Fig.
15a) and modeled outputs close to the surface (Figs. 15b and 15c), where the model output is
ammonium bisulfate and not just sulfate, and the actual mass concentrations of the models are
generally higher than the observations above 1 km (Fig. S1a). This sulfate-based compound
dominates the total aerosol mass for both the observations and model outputs, as is expected based
on its source air mass from East Asia, and accounts for the similar observed and modeled total
mass shape profiles (Fig. S1). The organic mass fraction, on the other hand, is lower by almost
half of the observed organic mass fraction due to sulfate dominating. This has a direct effect on
modeled aerosol hygroscopicity (kappa), as it has been shown earlier that f(RH) decreases with
increased organic mass fraction (Fig. 4).

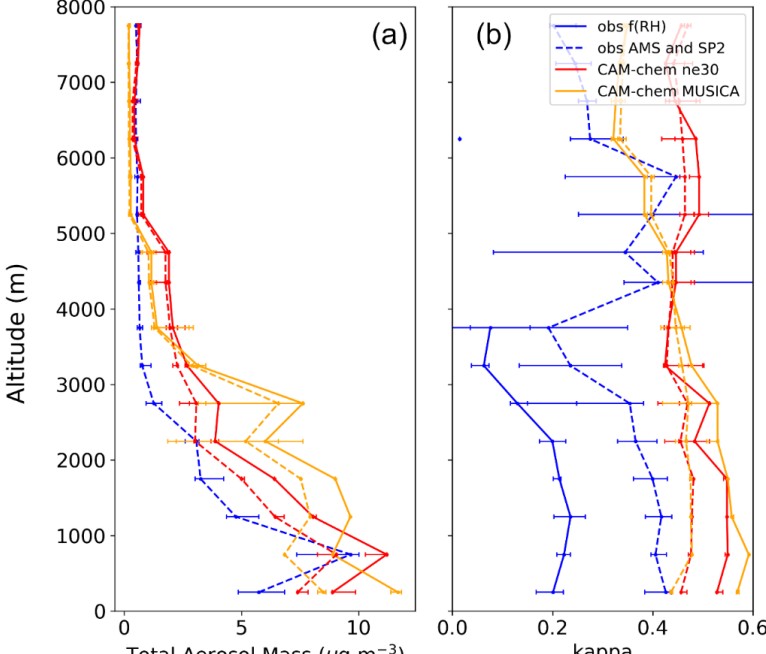


Figure 14. Vertical profiles of observed data (01:53 to 06:20 UTC) and CAM-chem model outputs (06:00 UTC) for
ne30 ~1° and MUSICA 0.25° grids for the tropical cyclone-induced convection case on 20 September 2019 at 500 m
intervals. (a) Total observed submicron aerosol mass from AMS (organics, $SO_4^{2-}$, $NO_3^-$, $NH_4^+$, $Cl^-$) and SP2 (black
carbon (BC)) data and CAM-chem output (organics: primary/hydrophobic, hygroscopic, and secondary ($C_{15}H_{38}O_2$),
sulfate ($NH_4HSO_4$), sea salt, dust ($AlSiO_5$), and black carbon (primary/hydrophobic and hygroscopic)) where the solid
line includes all the CAM-chem species and the dashed line excludes dust and sea salt and (b) calculated kappa from
observed f(RH) (solid line, < 5 µm) and from AMS and SP2 (dashed line, submicron) data and using ZSR mixing rule
for all CAM-chem aerosol species (solid line, submicron) and excluding dust and sea salt (dashed line). The lines
correspond to the median values of data in the given altitude intervals and the bars correspond to the 25$^{th}$ and 75$^{th}$
percentile values.

Median modeled kappa for the tropical cyclone-induced convection case is more than twice in
magnitude compared to the median derived kappa from f(RH) (Fig. 14b), though they have a
similar vertical profile shape. It has been noted from previous studies that the conversion of
observed f(RH) to kappa may be associated with up to 40% uncertainty (van Diedenhoven et al.,
2022), though this does not change that the kappa from CAM-chem is still approximately twice
the calculated observed kappa. Kappa calculated from f(RH) represents larger particles (< 5 µm)
compared to the kappa calculated from CAM-chem (< 0.48 µm), and the difference in size of the
particles could be contributing to this disparity in kappa.

We compare more similar sized particles, kappa derived from AMS and SP2 measurements (<
0.70 µm) with kappa calculated from CAM-chem (Fig. 14b) and find that the kappa derived from
the AMS and SP2 is still lower than that computed from CAM-chem outputs. The kappa from



AMS and SP2 is approximately twice (100% larger than) the kappa from f(RH), probably due to particle size differences. Based on Mie theory and studies comparing f(RH) of $PM_1$ and $PM_{10}$ particles (Zieger et al., 2013; Titos et al., 2021), $CAMP^2Ex$ f(RH) would be larger if it were measuring just submicron particles due to the increased scattering efficiency for accumulation mode particles compared to coarse particles. Though, based on Fig. 3 of Titos et al. (2021), the difference in f(RH) in marine sites would only be ~0.1 (given median observed SAE) and that would translate to ~20 to 40% increase in kappa. Thus, although size plays a role, composition is also contributing to the difference in the calculated kappa values between observations and the model.

The over and under-represented mass of sulfates and organics, respectively (Fig. 14b), by the model may be causing the higher kappa in the model. Based on the discussion in section 3.1.2 on air masses coming from East Asia (Fig. 4c), as is the case for this event, an organic mass fraction that is lower by half can increase the f(RH) (f(RH) = (-0.81 * organic mass fraction) + 1.56) and therefore the derived kappa by ~70%. Organics, especially secondary organic aerosols have been underpredicted by CAM-chem in urban and urban outflow regions (Schwantes et al., 2022). This is consistent with the observations for this case, which is influenced by urban outflow from East Asia and thus affecting the calculated kappa from the model.

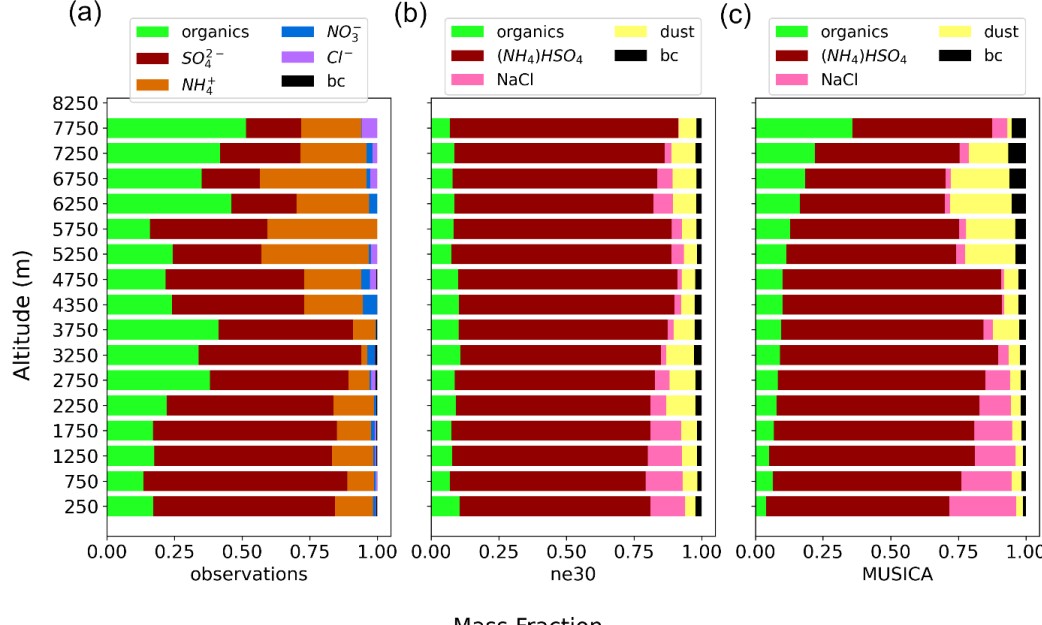

Figure 15. Vertical profiles of submicron median mass fractions at 500 m intervals of (a) observational data (01:53 to 06:20 UTC) and (b-c) corresponding CAM-chem model outputs (06:00 UTC) for (b) ne30 ~1° and (c) MUSICA 0.25° grids for the tropical cyclone-induced convection case on 20 September 2019. CAM-chem outputs were combined



into five main categories: organics (primary/hydrophobic, hygroscopic, and secondary ($C_{15}H_{38}O_2$)); sulfate
($NH_4HSO_4$); sea salt; dust ($AlSiO_5$); and black carbon (primary/hydrophobic and hygroscopic).
For the case of shallow convection, the shape of the vertical profile of the CAM-chem model
output total mass concentration is similar to the observed total mass concentration profile (Fig.
16a), though it underestimates the observed total concentration by approximately 2 to 4.5 µg m$^{-3}$
below 1 km altitude, and then overestimates concentrations by approximately 0.5 to 2 µg m$^{-3}$ above
1 km. This is due to the model underestimating organics and sulfate close to the surface, while
sulfate is overestimated above 1 km (Fig. S1b). The effect of this is seen in the model-derived
kappa, which is, like the previous case, higher than the calculated kappa from observations, though
to a lesser degree (Fig. 16b). Unlike the case of the tropical cyclone-induced convection, the
modeled kappa is relatively unchanging in shape compared to the kappa derived from
observations. This is likely due to the differences in the compositional profile at the higher
altitudes, even if the total mass is similar. The model is not able to represent the increase and
decrease in organic mass fraction from 3.5 to 4.5 km (Fig. 17a) and dominance of sulfate above
4.5 km. At those altitudes the observations become dominated by sulfate, ammonium, and nitrate
(Fig. 17a), even with losses due to scavenging and activation as shown by decreased aerosol mass
with altitude, probably due to the in-cloud production of sulfate as discussed in section 3.2.2. The
mass fraction profiles from the model (Fig. 17b) appear to have a steadily decreasing (increasing)
organic and sea salt (sulfate and dust) mass fraction with height. This case is a smaller scale event
with smaller surface winds associated with it (Fig. 13b) compared to the tropical cyclone case.
With weak forcing, it is likely that the CAM cumulus parameterizations does not predict the
convection observed in the shallow convection case affecting modeled aerosol mass vertical
distribution and, consequently, kappa in both the uniform ~1$\degree$ global model and the regionally
refined simulations. This is similar to another model evaluation study (i.e. GEOS and GOCART)
for the CAMP[2]Ex data, which recommended an improvement of shallow convection schemes to
improve the representation of vertical transport (Collow et al., 2022). In fact, the CO increase
above 5 km, that that we used to identify convection, is not captured by CAM-chem (Fig. S2).
What is consistent though is that for both the observations and model output of this shallow
convection case, the relative magnitude of the model kappa is lower than that of the tropical
cyclone case.

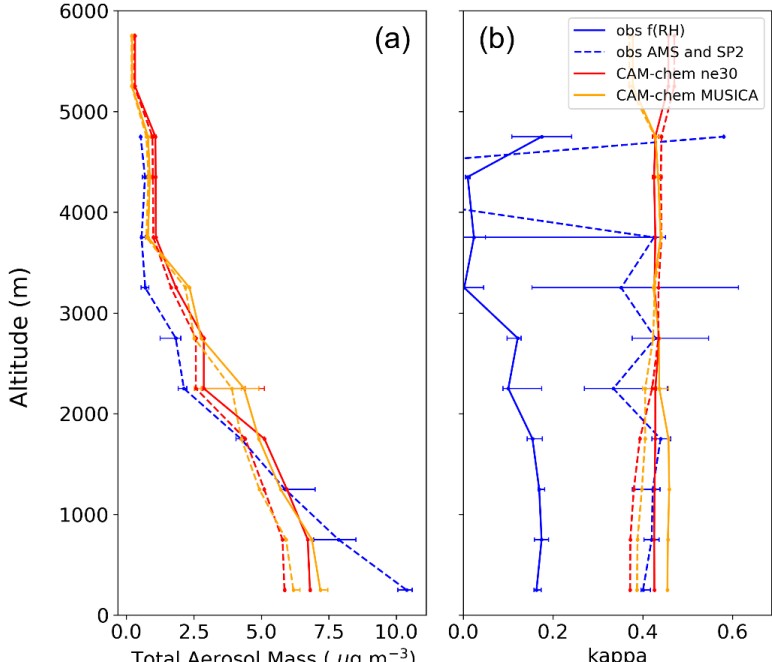

Figure 16. Vertical profiles of observed data (02:55 to 06:02 UTC) and CAM-chem model outputs (06:00 UTC) for ne30 ~1° and MUSICA 0.25° grids for the shallow convection case on 24 September 2019. (a) Total observed submicron aerosol mass from AMS (organics, $SO_4^{2-}$, $NO_3^-$, $NH_4^+$, $Cl^-$) and SP2 (black carbon (BC)) data and CAM-chem output (organics: primary/hydrophobic, hygroscopic, and secondary ($C_{15}H_{38}O_2$), sulfate ($NH_4HSO_4$), sea salt, dust ($AlSiO_5$), and black carbon (primary/hydrophobic and hygroscopic)) where the solid line includes all the CAM-chem species and the dashed line excludes dust and sea salt and (b) calculated kappa from observed f(RH) ) (solid line) and from AMS and SP2 (dashed line) data and using ZSR mixing rule for all CAM-chem aerosol species (solid line) and excluding dust and sea salt (dashed line). The lines correspond to the median values of data in the given altitude intervals and the bars correspond to the $25^{th}$ and $75^{th}$ percentile values.



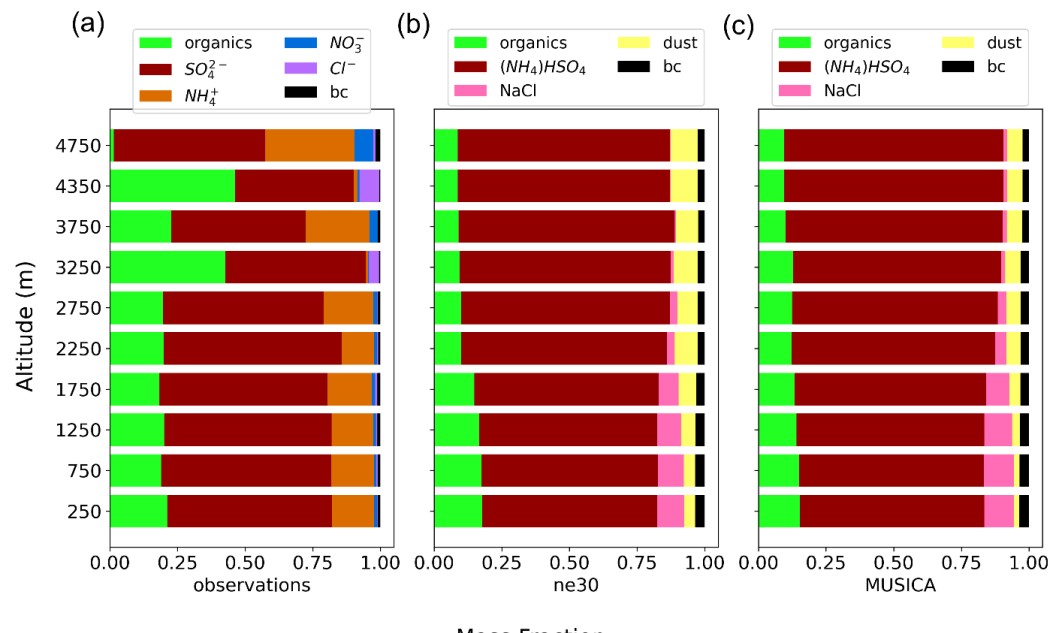

Figure 17. Vertical profiles of submicron mass fractions at 500 m intervals of (a) observational data (02:55 to 06:02 UTC) and (b-c) corresponding CAM-chem model outputs (06:00 UTC) for (b) ne30 ~1° and (c) MUSICA 0.25° grids for the shallow convection case on 24 September 2019. CAM-chem outputs were combined into five main categories: organics (primary/hydrophobic, hygroscopic, and secondary ($C_{15}H_{38}O_2$)); sulfate ($NH_4HSO_4$); sea salt; dust ($AlSiO_5$); and black carbon (primary/hydrophobic and hygroscopic).

**4. Conclusion**

This study reports on low $PM_5$ ($D_p$ < 5 µm) aerosol hygroscopicity measured during CAMP[2]Ex 2019 in Southeast Asia due to organics from biomass burning. Aging and vertical transport changes the hygroscopicity of particles affecting clouds and cloud representation in models in the region, emphasizing the need to improve emissions inventories and shallow cloud parameterizations. Notable results of this work, following the study goals raised at the end of Section 1 are as follows:

The generally low median f(RH) (1.24 with Q1 and Q3 of 1.05 and 1.43) of aerosol particles (< 5 µm) in Southeast Asia during the CAMP[2]Ex campaign from 24 August to 5 October in 2019 is linked to the dominating regional effect of biomass burning from the Maritime Continent. The median f(RH) of air masses from the Maritime Continent was exceptionally low (1.05 with Q1 and Q3 of 0.94 and 1.20). Measurements of f(RH) in other polluted marine environments around the world for $PM_{10}$ (< 10 µm) and $PM_1$ (< 1 µm) particles have higher ranges (median of ~1.7 – 2.0) of f(RH) values (Titos et al., 2021). The air masses with the highest f(RH) from CAMP[2]Ex are generally from the West Pacific, in the northernmost regions of the Philippines and farthest



away from the Maritime Continent. Their median f(RH) (1.49 with Q1 and Q3 of 1.26 and 1.73)
is also lower than what has been associated with typical polluted marine environments.
Throughout CAMP[2]Ex, submicron organic matter is the main aerosol component associated with
biomass burning and the low f(RH) values. Organics are a major feature of total aerosol mass,
especially in air masses traced back to the Maritime Continent. Biomass burning smoke is spread
out in the region contributing to elevated AOD downwind of the MC (Fig. 2 in Reid et al. (2023))
and is corroborated by cases of sub-1 f(RH) and submicron reflective particles with high organic
mass fractions that are present in all the air masses. Based on clustering analysis using optical
properties, majority of the particles sampled during the campaign appear to be a mixture of both
elemental and organic carbon. Organics dominate in terms of mass fraction due to the smoldering-
type burning in peat fire (Reid et al., 2023) from the Maritime Continent, which is known to
produce more organics than elemental carbon (Reid et al., 2005). This is consistent with what
Miller et al. (2023) observed where the highest median levels of organic and elemental carbon
mass during CAMP[2]Ex were due to biomass burning from the Maritime Continent. Without
organics, the baseline f(RH) (1.38 – 1.55, Fig. 4), is still relatively low, compared to measurements
from other areas (i.e. 2.19 from SEAC[4]RS in and around the U.S.) (Shingler et al., 2016a) probably
due to the presence of elemental carbon, which is the second most dominant aerosol type during
the campaign.
Farther away from the Maritime Continent, the organic particles have aged and become more
oxidized as they interacted with the other air masses. The West Pacific is a relatively remote region
in northern Philippines and downwind of biomass burning from MC, creating a combined marine
and polluted aerosol. The oldest biomass burning aerosols were observed there. The organic
oxidation values of the aerosol particles in the air mass from the West Pacific are close to the
threshold of maximum oxidation that has been observed in previous studies (Cubison et al., 2011).
In the West Pacific, vertical transport decreases and increases hygroscopicity at cloud level with
increased organic and sulfate mass fractions, respectively. These aerosol constituents are thought
to be transported from the MC and the Philippines and their surrounding oceans, where natural
and industrial sources are significant (Miller et al., 2023), and then cloud processed.
Evaluation of the global chemical transport CAM-chem model at two output grids (~111 km and
25 km) against two convective cases from CAMP[2]Ex show the underrepresentation of organics in
general and a higher derived aerosol hygroscopicity (kappa), which may be linked to the possible
overestimation of aerosol hygroscopicity from biomass burning from other studies in the area
(Collow et al., 2022; Edwards et al., 2022). CAM-chem overestimates sulfate in the tropical
cyclone induced convection case, consistent with results from an assessment of the NAAPS
reanalysis product on the positive bias of sulfate from East Asia (Edwards et al., 2022). The vertical
representation of the aerosol composition for the larger scale convection case due to a tropical
cyclone is better than that for the shallow convection case. Cloud processing and increased
hygroscopicity are not captured by the model for the shallow convection case, irrespective of
model grid size. The representation of shallow convection in the area by global models remains
challenging, based on similar model-evaluation studies (Collow et al., 2022). Past studies on
biomass burning aerosol effects on convection in Southeast Asia using a cloud-scale model



emphasized the importance of aerosol composition and absorptive properties, and their effect on
atmospheric stability, in the understanding of aerosol-cloud interactions (Hodzic and Duvel, 2018).
It is thus ideal that cloud-scale models be evaluated using the dataset from CAMP$^2$Ex, where the
invigoration of shallow clouds has been observed (Reid et al., 2023).
The implications of the low aerosol hygroscopicity in the region and its effects on clouds and
climate are just beginning to be unraveled. The mixing and aging of organic and elemental carbon
from biomass burning smoke in the Maritime Continent with background and transported sources
influences hygroscopicity observations and modeling uncertainties and can be the topic of future
work. Improvements in harmonization in terms of aerosol particle sizes and composition, along
with updated emissions inventories, will be helpful moving forward both for the observations and
modeling of aerosol hygroscopicity. These suggestions can hopefully improve shallow cumulus
representation, which is still the biggest source of differences in model sensitivity in the
understanding of climate change (Bony and Dufresne, 2005), in the region and globally. Emerging
endeavors to implement higher resolution schemes in a large domain (Pfister et al., 2020; Radtke
et al., 2021) to capture both fine scale aerosol-cloud processes along with improved observations
in Southeast Asia hold promise.

*Data availability.* CAMP$^2$Ex data can be found at
https://doi.org/10.5067/Suborbital/CAMP2EX2018/DATA001. CAM-chem model outputs can
be found at https://doi.org/10.6084/m9.figshare.26755936.v1.
*Author contributions.* LDZ, ECC, JPDG, GSD, RF, MRAH, MS, JW, QX, and AS carried out
various aspects of the data collection. ST and JZ conducted the CAM-chem model simulations.
GRL, LDZ, AFA, MB, ECC, RF, MRAH, MS, JW, QX, and AS performed analysis and
interpretation of the data. GRL and AS prepared the manuscript with contributions from the
coauthors.
*Competing interests.* At least one of the (co-)authors is a member of the editorial board of
*Atmospheric Chemistry and Physics.*
*Special issue statement.* This article is part of the special issue "Cloud Aerosol and Monsoon
Processes Philippines Experiment (CAMP$^2$Ex) (ACP/AMT inter-journal SI)". It is not associated
with a conference.
*Acknowledgements.* The authors gratefully acknowledge the NOAA Air Resources Laboratory
(ARL) for the provision of the HYSPLIT transport and dispersion model and READY website
(https://www.ready.noaa.gov) used in this publication. The NSF National Center for
Atmospheric Research is a major facility sponsored by the U.S. National Science Foundation
under Cooperative Agreement No. 1852977.
*Financial support.* This research has been supported by the National Aeronautics and Space
Administration (grant no. 80NSSC18K0148) and ONR grant N00014-21-1-2115.



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
