# Peer review of "Measurement Report: Characterization of Aerosol"

_EGUsphere, 2024_

## Author Response (AR1)

*REVIEWER # 1*

*Response: We thank reviewer # 1 for the comments that have helped improve our manuscript and highlight its value. Below we provide responses to referee comments and suggestions in italicized bold font. All changes to the manuscript can be identified in the version submitted using Track Changes.*

Review of "Measurement Report: Characterization of Aerosol Hygroscopicity over Southeast Asia during the NASA CAMP2Ex Campaign" by Lorenzo et al.

Anonymous Referee #1

November 1, 2024

This report details a study of the spatial distribution of aerosol hygroscopicity in Southeast Asia during NASA's "CAMP2Ex" mission, a monsoon impacted region / period. Hygroscopicity is discussed in the context of the overall aerosol composition and likely sources, e.g. regional biomass burning. The results are then evaluated using a chemical transport model (CAM-chem) to show that, compared to the data, there is an underrepresentation of organics resulting in overestimated modeled aerosol hygroscopicity.

Overall, this is a solid "Measurement Report" with important information that should appear in a peer-reviewed publication as well as relevant scientific results. Pending minor revisions, it can be published in ACP.

*Response: We thank the reviewer for the appreciation of our work and are encouraged by the comments and suggestions which we have responded to in detail below.*

Main Paper

Introduction –

The introduction is well written with an extensive treatment of kappa. I believe this section would benefit from relevant values of kappa for the aerosol types being discussed (i.e., marine, organic, EC are mentioned and ranked comparatively, can you add and reference their kappa values? – e.g. those found in the supplement?).

*Response: A table of hygroscopicity values from selected summary papers was added to the introduction. And the text was edited to include referencing to Table 1. "Observed and simulated aerosol hygroscopicity using aforementioned parameters are greater (Table 1) in clean marine air masses compared to air masses over land, near terrestrial biogenic sources which are secondary organics precursors, and under polluted conditions (Swietlicki et al.,*

*2008; Duplissy et al., 2011; Petters and Kreidenweis, 2007; Burgos et al., 2019; Titos et al., 2021)."*

*Table 1: Hygroscopicity values of various aerosol types based on selected summary studies for measured g(RH) (Swietlicki et al., 2008) and f(RH) (Burgos et al., 2019 and Titos et al., 2021) and modeled κ (Pringle et al., 2010).*

| Aerosol Type | g(RH=90%) (10 - 500 nm) | f(RH=85%/RH=40%) (< 1000 nm) | κ (5 - 500 nm) |
|---|---|---|---|
| Clean Marine | 1.62 - 2.14 | 2.1 - 2.3 | 0.92 |
| Polluted Marine | 1.00 – 1.76 | 1.6 - 2.2 | 0.59 |
| Background Continental | 1.00 - 1.60 | 1.29 - 1.4 | 0.15 - 0.17 |
| Urban Continental | 1.00 - 1.68 | 1.38 - 2.1 | 0.21 - 0.36 |
| Aged Biomass | 1.15 - 1.30 | 1.02 - 2.1* | < 0.1 |
| New Particle Formation | 1.04 - 1.40 | - | - |
| Free Troposphere | 1.21 - 1.47 | - | site specific |
| Arctic | 1.19 - 1.44 | 2.7 | ~0.4 |

*\*Gomez et al., 2018 (f(RH=85%/RH=25%))*

*The text was also edited such that a range of kappa values for organics, elemental carbon, and inorganics based on Table S1 were added. "Continental aerosol particles have smaller diameters and are usually less hygroscopic due to more organic-rich aerosol particles (κ = 0 - 0.1 from Table S1) and pure elemental carbon (EC, κ = 0 - 0.035 from Table S1) particles (Wang et al., 2014; Kreidenweis and Asa-Awuku, 2014). Organics are generally less hygroscopic than inorganics (κ = 0.56 – 1.24 from Table S1), and their hygroscopicity is affected by oxidation level (e.g., O:C ratio), oxidation state, and solubility (Brock et al., 2016a; Wu et al., 2016; Thalman et al., 2017)."*

The paragraph starting at line 126 needs to be rewritten. "In the CAMP2Ex region, biomass burning aerosol hygroscopicity is over-estimated by global atmosphere models simulating the CAMP2Ex campaign…"

*Response: The first sentence was rewritten. And another sentence added "Global atmospheric model evaluation of aerosol optical depth and extinction for the CAMP[2]Ex region suggests that biomass burning aerosol hygroscopicity is over-estimated by global atmosphere models, with both the model size representation of the aerosol particles and the size discrepancy between model and observations contributing to this (Collow et al., 2022; Edwards et al., 2022). The findings from Collow et al. (2022) and Edwards et al. (2022) bring up the need for a more thorough understanding of the components of biomass burning aerosols in the region that affect observed and modeled hygroscopicity."*

Note that the superscript 2 is sometimes used and sometimes not used (please be consistent).

*Response: Superscript has been applied to all 2's that appear as part of the CAMP²Ex acronym.*

The phrasing here seems to indicate the Collow et al. and Edwards et al. references have already done what this paper claims to do (next paragraph "To our knowledge, this is the first time this dataset has been explored extensively to characterize aerosol hygroscopicity properties in the region." – please clarify what of the points i – iv has been done and what is novel about this report? Specifically, how are i and ii different from what is done in the above references?

*Response: A sentence was added after line 96 noting that there has been no extensive analysis of aerosol hygroscopicity in our region of interest. The underlined text refers to the edited text. "Although aerosol studies in the rapidly developing Southeast Asia (SEA) region are increasing, few are focused on the nature of aerosol particles and their interactions with water vapor and clouds (Tsay et al., 2013; Ross et al., 2018; Reid et al., 2023). There has been no comprehensive study of aerosol hygroscopicity in and around the Philippines in Southeast Asia."*

*As noted in an earlier response, a sentence was added to the previous paragraph (line 136) to show how work is a natural next step based on the work of Collow et al. and Edwards et al. "The findings from Collow et al. (2022) and Edwards et al. (2022) bring up the need for a more thorough understanding of the components of biomass burning aerosols in the region that affect observed and modeled hygroscopicity."*

*The conclusion was also updated to differentiate this work from Collow et al. (2022) and Edwards et al. (2022). "Evaluation of the global chemical transport CAM-chem model at two output grids (~111 km and 25 km) against two convective cases from CAMP2Ex show the underrepresentation of organics in general and a higher derived aerosol hygroscopicity (kappa), which may be linked to the possible overestimation of aerosol hygroscopicity from biomass burning based on studies of aerosol optical depth and extinction in the area (Collow et al., 2022; Edwards et al., 2022)."*

*The statement in point iv was also updated to include the regionalization capability of CAM-chem: "(iv) evaluate a global chemical transport model with regional refinement for aerosol vertical transport"*

At line 139 the word "opportune" seems unnecessary, please remove.

*Response: The word "opportune" has been deleted.*

Methods –

At line 162 and elsewhere "submicron" should be "submicrometer"

*Response: The word "submicron" was replaced with "submicrometer" for the 35 instances that it was used in the text.*

At 2.3.1 Trajectory Analysis – Can you include some detail on the trajectory results? Direct reference to Hilario is fine but some detail on how the trajectories are obtained should be contained in the paper; this is probably more a description of HYSPLIT than anything but, as this is directly impactful on this report, please make the reader aware of the method (i.e., as is done for CAM-chem).

*Response: HYSPLIT was mentioned in the first sentence of the section after citing Hilario et al. Information about the five-day trajectories derived from 1-minute flight locations and the input meteorology were added as noted in the edited text copied below. The underlined text refers to the edited text.*

*"This work leverages trajectory results explained in detail by Hilario et al. (2021) using the National Oceanic and Atmospheric Administration Hybrid Single Particle Lagrangian Integrated Trajectory Model (HYSPLIT) (Stein et al., 2015; Rolph et al., 2017). Five-day HYSPLIT back trajectories from specific CAMP²Ex flight locations were associated with air masses undergoing long-range transport nearby from either the Maritime Continent (MC), East Asia (EA), peninsular Southeast Asia (PSEA), or the West Pacific (WP) when they were within the regions at altitudes below 2 km for more than 6 h. The back trajectories were generated based on 1-minute resolution aircraft locations and 0.25° x 0.25° resolution NOAA Global Forecasting System (GFS) archived meteorological data."*

Results and Discussion

Throughout this section (and through the Conclusions) subjective terms such as relatively low, narrow range, etc. are used. Please rewrite this as quantities and remove subjective terminology (as an example, f(RH) being relatively low needs to be compared to something for context, better instead to simply reported the observed values).

*Response: The text with the subjective terms was edited as noted below, where those parts that are underlined were edited.*

*Abstract: "Median f(RH) is low (1.26 with lower to upper quartiles of 1.05 to 1.43) like polluted environments, due to the dominance of the mixture of organic carbon and elemental carbon."*

*Section 3.1.1 1ˢᵗ paragraph: "The f(RH) values in the CAMP²Ex campaign (Fig. 1a) had an interquartile range of values between 1.05 and 1.42 and were low (median of 1.24 for 143,107 1 s points) compared to other polluted marine environments (Titos et al., 2021)."*

*Section 3.1.1 2$^{nd}$ paragraph: "The air mass from EA (Fig. 2a) has the narrowest range of values (Q1: 1.28, median (Q2): 1.38, Q3: 1.47) compared to other air masses, likely representative of urban aerosol particles."*

*Section 3.1.2 4$^{th}$ paragraph: "The presence of coarser particles from the WP, based on its marine origin (with higher DMA mass fraction, Fig. 3b) and high sea salt fraction (54%, Fig. 3b), contributed to it having the highest median f(RH) amongst the air masses."*

*Section 3.1.2 7$^{th}$ paragraph: "Compositional data sheds some light on this, because the particles classified as EC dominated have higher submicrometer sulfate (0.43) and bulk sea salt mass (0.38) fractions (Fig. 5b) compared to particles classified as OC (0.16 and 0.22) and dust dominated (0.35 and 0.19), which are known to have low hygroscopicity in general."*

*Section 3.2.3 4$^{th}$ paragraph: "This could explain the higher total mass values from the model outputs compared to observations for some altitude levels, though in general even without dust and sea salt the CAM-chem model still has slightly higher aerosol total mass values (dashed lines in Fig. 13a) compared to the observations."*

*Section 3.2.3 4$^{th}$ paragraph: "The CAM-chem model represents the approximate aerosol total mass well for this case probably because of its large-scale nature, where wind speeds were high (> 20 m s$^{-1}$) and where the aerosol particles are potentially from a large source (East Asia) (Figs. 9a and 13a)."*

*Conclusions: "The generally low median f(RH) (1.24 with Q1 and Q3 of 1.05 and 1.43, respectively) of aerosol particles (< 5 μm) in Southeast Asia during the CAMP$^2$Ex campaign from 24 August to 5 October in 2019 is linked to the dominating regional effect of biomass burning from the Maritime Continent. Measurements of f(RH) in other polluted marine environments around the world for PM$_{10}$ (< 10 μm) and PM$_1$ (< 1 μm) particles have higher ranges (median of ~1.7 – 2.0) of f(RH) values (Titos et al., 2021)."*

*Conclusions: "The median f(RH) of air masses from the Maritime Continent was exceptionally low (1.05 with Q1 and Q3 of 0.94 and 1.20, respectively) for polluted marine environments but are more similar to those with biomass burning (Gomez et al., 2018)."*

*Conclusions: "Biomass burning smoke is spread out in the region increasing AOD downwind of the MC (Fig. 2 in Reid et al. (2023)) and is corroborated by cases of sub-1 f(RH) and submicrometer reflective particles with elevated organic mass fractions that are present in all the air masses."*

*Conclusions: "Without organics, the baseline f(RH) (1.38 – 1.55, Fig. 4), is still lower than measurements from other areas (i.e. 2.19 from SEAC$^4$RS in and around the U.S.) (Shingler et al., 2016a) probably due to the presence of elemental carbon, which is the second most dominant aerosol type during the campaign."*

*Conclusions: "The West Pacific is a remote region in northern Philippines and downwind of biomass burning from MC, creating a combined marine and polluted aerosol."*

One final statement is that this paper is on the long side and contains strange formatting. The paper would benefit from an extensive read through in an attempt to shorten; this is nothing especially troubling but, at 33 pages with 17 figures, the authors could consider if anything can be eliminated / combined (e.g. the vertical profile figures or combining case study figures)? I suspect the formatting is a result of some figure formats and will probably be corrected in final edits. None the less, there exist large gaps and occasional blank lines that should be corrected.

*Response: The observational plots for the case studies in Figures 9 and 11 were combined (now Figure 9) and the text references to the figures were updated. The model evaluation plots for the case studies in Figures 14 to 17 were combined (now Figure 13) and the text references to the figures were updated. Unnecessary spaces and blank lines were removed. The manuscript text font was updated based on the ACP template (https://www.atmospheric-chemistry-and-physics.net/Copernicus_Word_template.docx). The manuscript text was decreased to 10 and the spacing increased to 1.5 as noted in the template and the manuscript is now down to 30 pages with 13 figures.*

*REVIEWER # 2*

*Response: We thank reviewer # 2 for the appreciation of our work, motivating comments, and clarifications, which helped to improve the communication of our study. We are thankful for the reviewer's thorough analysis of our methods and analysis, and we have incorporated the reviewer's suggestions as best as we could. We provide responses to referee comments and suggestions in italicized bold font. All changes to the manuscript can be identified in the version submitted using Track Changes.*

Review of "Measurement Report: Characterization of Aerosol Hygroscopicity over Southeast Asia during the NASA CAMP2Ex Campaign" by Lorenzo et al.

Anonymous Referee #2

December 1, 2024

This study uses aircraft-based measurements of the aerosol light scattering humidity enhancement factor f(RH) made during the August-October 2019 NASA Cloud, Aerosol, and Monsoon Processes Philippines Experiment (CAMP2Ex field campaign over SE Asia. The CAMP2Ex was designed to understand the role of aerosol particles in cloud formation and in regulating solar radiation during the southwest monsoon. The goals of this proposed study are to (i) characterize the spatial distribution of aerosol hygroscopicity in Southeast Asia during the CAMP2Ex airborne mission, (ii) relate aerosol hygroscopicity and composition, (iii) identify emission events that impact aerosol hygroscopic growth, and (iv) evaluate a global chemical transport model for aerosol vertical transport

The authors provide an excellent discussion of background, including the importance of aerosol hygroscopicity measurements. The measurements and methods are well-described, although I have a few suggestions for additions (See below). The authors use particle light scattering at 2 RH values (< 40% and ~82%) to calculate the $\gamma$ fit parameter (Eq. 3 of Titos et al., 2021 and many other papers) of the f(RH) curve , which is then used to calculate f(RH) as the ratio of scattering at RH=80% divided by that at RH=20%. The f(RH) was then used to calculate the optical kappa factor $\kappa_{opt}$, using Eq. 5 of Brock et al (2016). For comparisons with CAM-modeled $\kappa_{chem}$ and with $\kappa_{chem}$ derived from collocated aerosol chemical speciation measurements, the authors used the relationship $\kappa_{chem} = 0.56 / \kappa_{opt}$ (Brock et al., 2016). Median f(RH) during the field campaign was lower than those from other studies and the $\kappa_{chem}$ derived from measurement-based f(RH) was lower than that predicted by the models by a factor of ~2 (Fig. 14). Median f(RH) <1 was reported for the "organic-dominated" aerosol type (Fig. 5). The authors also calculate intensive AOPs such as SSA, SAE, AAE for cases when scattering coefficient and absorption coefficient exceeds 2Mm-1. This lower threshold for scattering coefficient is very low (many of the papers based on NOAA FAN data use 5Mm-1 as a lower threshold for scattering coefficient).

The paper is well-written and the collocated aerosol chemical speciation and CO measurements, along with HYSPLIT back-trajectories, facilitate a thorough investigation of source region and aerosol composition influences on measured f(RH) during the campaign. I recommend that it be published after consideration of the suggestions below, particularly those related to the apparent under-estimation of f(RH).

**Suggestions:**

**(1) Possible role of measurement/analysis methods in the low f(RH) values**: It is conceivable that experimental conditions and the way in which f(RH) was calculated could have played a role in f(RH) values that were lower than other studies and the f(RH)<1 values. Below I briefly elaborate on 4 sources that could potentially lead to inaccurate f(RH) values. At the very least, these should be discussed in the paper.

- The authors did a nice job of studying possible reasons for f(RH) lower than reported in other field campaigns, including the relationships between low f(RH) and high OA fractions. However, the way in which f(RH) was calculated (based on only two RH values to calculate γ, as I summarized above) could also have contributed to the low f(RH). Most studies (Burgos, et al. 2019 and others) use the full humidogram scans for the γ fit, which is better because the slope of wet/dry scattering curve isn't constant and often increases near RH =>80%. I realize that a full humidogram scan is not feasible for the aircraft-based measurements but believe that the authors should at least consider this possible error source via a simple simulation or other means.

  *Response: Thank you very much for acknowledging our efforts in the analysis. The reviewer is correct that humidograms are not feasible for aircraft-based measurements, and that is why two parallel nephelometers were used during the campaign. Even then, the scattering RH-dependence would have to diverge significantly from a gamma relationship for measurements at 40% and 85% to not at least reasonably represent the humidity dependence. This is especially true for f(RH)<1 values, where a bad gamma fit cannot be the reason that scattering at 85% is less than scattering at 40%. We felt this comment did not have a natural way of entering the text of the manuscript and could be viewed as a slight distraction to the flow of the text for a paper that already can be considered to be quite long.*

- The authors briefly described how the RH probes were calibrated but a single-point calibration isn't always adequate for the Vaisala probes. Using a fit based on 2-3 saturated salts (including one at low RH to get the intercept of the calibration curve) can reduce the RH uncertainty by ~2-3% relative to a single point calibration, based on my experience. The authors also did not say whether the TSI nephelometer internal RH sensor was used

for the wet scattering RH value or whether a Vaisala RH probe placed near the exhaust was used, along with assumption of constant dewpoint. The latter method is often preferred due to the large RH errors (up to ~10%) in the Honeywell RH sensors placed in TSI nephelometers. Part of this error can arise due to the sensor itself while part is due to the location of the sensor inside the nephelometer. The authors should explicitly state which RH sensor (nephelometer internal sensor or other) was used and how the sensor was calibrated (i.e. with sensor taken out of the nephelometer or left in its normal sampling location) and consider this as a possible source of systematic error in f(RH). For example, an RH sensor reading that is even a few percent too high can lead to underestimation of f(RH).

*Response: We have updated (underlined text) the 2ⁿᵈ paragraph in 2.2.2 to explicitly state that internal sensors were used because of their relevant placement for sampling. We also describe in detail how the RH sensors were calibrated from ~60% -90% RH. Uncertainties at low RH are stated, with the note about less sensitive f(RH) calculations at lower RH.*

*"Each nephelometer was calibrated with pure $CO_2$ prior to the campaign and zero checks were performed periodically during the flights to prevent baseline drift. Hydrophilic polystyrene latex (PSL) aerosol were also introduced periodically in-flight to ensure the nephelometers were consistent. All nephelometer scattering coefficient measurements were corrected for truncation errors using a documented method (Anderson and Ogren, 1998). System response was verified in flight by introducing hydrophobic PSL spheres into the sample stream to ensure an f(RH) value of 1.0 is observed. Relative humidity measurements for the calculation of f(RH) were calibrated in the laboratory using nebulized monodisperse 200 nm ammonium sulfate to ensure deliquescence at ~80% RH at the typical measurement flow rate (i.e. 10 L min⁻¹) (Brooks et al., 2002). The RH was slowly ramped up from ~60% – 90% and the apparent deliquescence RH was calculated based on the sharp increase in scattering in the humidogram. The offset between the calculated RH and the deliquescence RH for ammonium sulfate (i.e., ~79.5%) was applied to the data post-processing. Calibration at ~80% RH increases the confidence in the humidified nephelometer data accuracy. Non-linearities can occur at low RH measurements and add to uncertainties, but the f(RH) calculation is much less sensitive at the dry end of the spectrum. Internal RH sensors were used because their placement is most relevant to the sampling condition inside the nephelometer.  Note that sampling efficiency decreases for supermicrometer diameter particles with increasing size up to the 5-μm inlet cutoff, due to losses in transport tubing and in the drying/humidification control system. Thus, derived f(RH) is applicable to accumulation-mode particles and is partially sensitive to coarse-mode particles from 1-5 μm diameter."*

- Comparisons of f(RH) with those reported in other studies (ex: Titos, et al., 2021) must consider the RH values used, since f(RH) is dependent on RH. Burgos et al., (2019), Titos et al (2021), and most other studies used RH=85% and 40% to calculate f(RH) while the current study uses RH=80% and RH=20% to compute f(RH). As such, the f(RH) values from the two studies are not directly comparable.

*Response: We added a table (Table 1 in the text and shown below) of the hygroscopicity values from other studies in the introduction and include the wet and dry RH values that were used. We updated the text in the first paragraph of section 2.2.2 to note the dry and wet relative humidity levels used in the parallel nephelometers. We also added a sentence in the discussion of results in section 3.1.1 to note the slight differences in RH used for the different studies.*

*Table 1: Hygroscopicity values of various aerosol types based on selected summary studies for measured g(RH) (Swietlicki et al., 2008) and f(RH) (Burgos et al., 2019 and Titos et al., 2021) and modeled $\kappa$ (Pringle et al., 2010).*

| Aerosol Type | g(RH=90%) (10 - 500 nm) | f(RH=85%/RH=40%) (< 1000 nm) | $\kappa$ (5 - 500 nm) |
|---|---|---|---|
| Clean Marine | 1.62 - 2.14 | 2.1 - 2.3 | 0.92 |
| Polluted Marine | 1.00 - 1.76 | 1.6 - 2.2 | 0.59 |
| Background Continental | 1.00 - 1.60 | 1.29 - 1.4 | 0.15 - 0.17 |
| Urban Continental | 1.00 - 1.68 | 1.38 - 2.1 | 0.21 - 0.36 |
| Aged Biomass | 1.15 - 1.30 | 1.02 - 2.1* | < 0.1 |
| New Particle Formation | 1.04 - 1.40 | - | - |
| Free Troposphere | 1.21 - 1.47 | - | site specific |
| Arctic | 1.19 - 1.44 | 2.7 | ~0.4 |

*\*Gomez et al., 2018 (f(RH=85%/RH=25%))*

*The current study uses a dry RH of < 40% and a humidified RH(controlled to 82 ± 10%. The text was updated (text that has a strikethrough was removed). "The f(RH) parameter is calculated from the empirically derived exponential fit coefficient gamma, (γ),  (Ziemba et al., 2013). The gamma parameter is based on measurements of scattering at 550 nm at two different relative humidity levels: dry (< 40%) and humidified (controlled to 82 ± 10%)."*

*The following text was added after the 1st sentence in section 3.1.1: "We note that the comparisons to other studies may be affected by the slight difference in relative humidity (f(RH=85%/RH=40% for Titos et al. (2021) and f(RH=82 ± 10%/RH<40% for this study) used to calculate f(RH)."*

- The authors did not say whether both nephelometers were calibrated (zero and span checks) before and after the field campaign (my guess is that they were) and whether the scattering measured by both nephelometers at low RH was compared before the campaign.

*Response: The text was updated (underlined) in the 2nd paragraph of section 2.2.2 to state the procedures applied to the nephelometers.*

*"Each nephelometer was calibrated with pure $CO_2$ prior to the campaign and zero checks were performed periodically during the flights to prevent baseline drift. Hydrophilic polystyrene latex (PSL) aerosol were also introduced periodically in-flight to ensure the nephelometers were consistent. All nephelometer scattering coefficient measurements were corrected for truncation errors using a documented method (Anderson and Ogren, 1998)."*

As a side note, I am surprised that this field campaign did not utilize f(RH) measurements at a ground site (along with dry particle scattering and absorption) to compare with those measured on the aircraft, especially given the difficulties in making such measurements. Comparisons of aircraft and ground-based f(RH) measurements would have provided valuable information as to the possible experimental role in the low f(RH) values.

*Response: It would have been ideal to have f(RH) measurements at a ground site to compare with those measured on the aircraft. We instead added some sentences to the discussion on kappa about past studies in the region in the 6th paragraph of section 3.2.3.*

*"Examples of studies in the region with low kappa values suggest that both the size and compositional effect together are important. Lab experiments of tropical peat burning showed that smaller (70 nm) primary organic particles (POA) became more oxidized faster compared to larger particles (150 nm) and kappa for oxidized POA (100 nm) was 0.16 (Chen et al., 2022). Ground measurements of aerosol particles (56 nm – 100 nm) in Quezon City in the Philippines, which were dominated by elemental carbon, yielded kappa values less than 0.10 (AzadiAghdam et al., 2019)."*

**(2) Cazorla (2013) Aerosol Classification Method**: I am not convinced that the Cazorla (2013) aerosol type classifications add much to the paper, unless the objective is to show that they don't work very well for the region and the time period studied. The paper is rather long and contains many figures (17) and could be shortened by eliminating this discussion, especially since aerosol chemical speciation and CO measurements are available.

*Response: We found that the Cazorla aerosol type classification to be helpful to confirm some of the analysis we have done. For example, even while AAE and SAE were calculated for < 5 μm particles, 93% of the data were submicron carbonaceous particles, which may explain the low f(RH) for < 5 μm particles measured during the campaign. We added a sentence (underlined) to state this to the 7th paragraph of section 3.1.2.*

*"Most of the aerosol particles (< 5 μm dry diameter and 93% of all particles) collected during the CAMP²Ex field campaign have optical properties (Fig. 5a) that resembled EC/OC mix + mix (in this case we combined the sub-categories of EC/OC mix and mix) (51%) and EC dominated (42%) aerosol types. These carbonaceous particles are less hygroscopic, and their dominance helps to explain the low median f(RH) in the CAMP²Ex region."*

*Coarse particle and sea salt data, for example, were not available for AMS, and with the aerosol typing and PILS data we were able to verify that the coated large particles, which had the highest f(RH), were most probably marine in origin. A phrase was added (underlined) to the 6th paragraph of section 3.1.2 to note that there were not too many particles that were classified as coated large particles.*

*"The high f(RH) for the coated large particles is consistent with its the largest bulk sea salt mass fraction (0.56) (Fig. 5b) compared to other aerosol types, though they only make up 1.8% of the total number of particles."*

*The dust dominated aerosol type also made sense because it had the highest non-sea salt calcium from the PILS data. A sentence was added to the 6th paragraph of section 3.1.2 to note this.*

*"Dust dominated particles make up only 0.05% of the total number of particles and had the highest non-sea salt calcium and a median f(RH) of 1.15 (with Q1 and Q3 of 0.94 and 1.43)."*

*One of the distinct results from the aerosol typing was the association of sulfate to EC dominated aerosol particles, which could be from local shipping sources, East Asia, and the Maritime Continent. We added a phrase (underlined) to the 7th paragraph of section 3.1.2 to emphasize the importance of sulfate in peat burning emissions.*

*"Peat smoke particles from MC have also been found in past studies to have sulfate mixed with carbonaceous species (Nakajima et al., 1999), up to ~20% of the mass fraction of secondary organic aerosol from lab experiments on peat burning was attribute to sulfate (Chen et al., 2022)."*

*We appreciate the note about the paper length. As such, we removed the discussion on EC/OC mix + mix type and OC/dust mix. This is denoted by a strikethrough over the text as seen below.*

*"The presence of OC (0.64 submicrometer mass fraction) decreased the median f(RH) values (Fig. 5b) where the median f(RH) for the EC/OC mix + mix type was 1.03 (with Q1 and Q3 of 0.94 and 1.18)."*

*"As such, there are no PILS compositional data available for the OC/dust mix aerosol type. However, from available data from dust dominated particles, it is also possible that hygroscopic particles like ammonium and nitrate, which have the greatest bulk mass fractions (0.45 and 0.17) in the dust dominated particles compared to other aerosol types, partitioned to dust and increased f(RH) for the OC/dust mix particles."*

*Additionally, in order to decrease the number of pages further, we reduced the total number of figures. The observational plots for the case studies in Figures 9 and 11 were combined (now Figure 9). The model evaluation plots for the case studies in Figures 14 to 17 also were combined (now Figure 13).*

(3) AAE values less than one (as reported in many cases of this study) sometimes result from PSAP filter loading nonlinearity effects, especially in regions with high OA fractions. They can also result from low signal-to-noise during low aerosol loading conditions. The authors used a lower threshold of 2 Mm-1 for absorption coefficient data used in the study.

*Response: The threshold of 2 Mm$^{-1}$ was used to try to minimize noise in the calculation of AAE for 1 s data. The PSAP filters were changed to keep transmission above 70%. This standard practice minimizes non-linearity effects due to filter loading. The text in the 3$^{rd}$ paragraph of section 2.2.2 was updated with a phrase (underlined) and two additional sentences as found below.*

*"the absorption Ångström exponent (AAE, 470-660 nm) was computed when the absorption coefficient was greater than 2 Mm$^{-1}$ to try to minimize noise issues (Mason et al., 2018)."*

*"The PSAP filters were changed before transmission dropped below 70%. This standard practice minimizes non-linearity effects due to filter loading."*

(4) Median values of $\kappa_{chem}$ derived from measurement-based f(RH) were lower than those predicted by the models by a factor of ~2-3 (Fig. 14) and lower than median $\kappa_{chem}$ calculated from collocated AMS and SP2 measurements by a factor of ~2 . The authors used the relationship $\kappa_{chem}$ =0.56 / $\kappa_{opt}$ (Brock et al., 2016) but this relationship was derived from measurements in the SE US during a period (Summer 2013) where biogenic SOA was the dominant aerosol mass component and without much biomass burning influence. It is possible that the numerical constant of proportionality used by Brock may not work well in SE Asia during the CAMP2Ex field campaign but the authors did not consider this as a potential source for the large discrepancy (Fig. 14).

*Response: We updated the text in the 5th paragraph of section 3.2.3 to note that the southeast U.S. data used as a basis for the constant of proportionality had slightly smaller particles compared to CAMP2Ex. We also noted that Brock et al. (2016b) based their conversion on submicron f(RH) while in CAMP2Ex larger particles (< 5 μm) were included in the calculation of f(RH). The constant of proportionality could indeed be contributing to the discrepancy between modeled and observed kappa.*

*"We note that additional uncertainties as a result of the conversion Brock et al. (2016b) could also be contributing to the discrepancy. Brock et al. (2016b) conducted submicrometer measurements in the southeast U.S. while CAMP2Ex f(RH) was calculated for particles less than 5 μm."*

**Minor Suggestions**

(5) Fig 1(a) is confusing to me. It shows color coded f(RH) values overlaid on a lat/long plot with source regions labeled. However, I am not sure what I am looking at. Do each of the color coded datapoints correspond to the location of the measured f(RH) along the flight path or the "location" of the source region, based on the HYSPLIT trajectories and the methods outlined in Sect. 2.3.1? Maybe I am missing something obvious but a bit more clarification in the figure caption would be helpful.

*Response: Each of the color-coded data points correspond to the location of the measured f(RH). The figure caption (Fig. 1) has been edited to express this more clearly. "Map showing flight locations during CAMP2Ex color coded by f(RH) 1 s values and approximate locations of the air mass sources (black text) identified by Hilario et al. (2021) that affected CAMP2Ex."*

(6) The authors make the claim (bottom of P. 14) that "the highest median f(RH) values are from the WP and EA air masses (Fig. 2a). Both have lower organic mass fractions (0.29 and 0.30, respectively), but have distinctly larger aerosol size profiles based on their SAE values (Fig. 3a) ". The plot in Fig 3(a) supports this assertion for the WP air mass (whose SAE percentiles all are clearly lower than those of other source regions) but not for the EA air mass, whose SAE is not much different than those for the MC, PSEA, and Other source regions. Values of SAE are close to 2 for each of these source regions (Fig.3a) and indicates primarily accumulation-mode aerosols. Since the SAE is more sensitive to course and fine mode fractions and less sensitive to small differences in accumulation mode particle size (except for monomodal aerosols), it is difficult to make the claim that EA air masses have "distinctly larger aerosol size profiles" than MC, PSEA, and Other regions. This is especially true for lower values of $\sigma_{sp}$ (Authors use $\sigma_{sp} > 2$ Mm-1 as threshold for SAE calculations), due to larger SAE uncertainties.

*Response: The note about larger aerosol size profiles has been deleted for EA. The text in the 4th paragraph of section 3.1.2 has been updated accordingly. The edited texts are underlined.*

*"The highest median f(RH) values are from the WP and EA air masses (Fig. 2a). Both have lower  organic mass fractions (0.29 and 0.30, respectively) compared to other air masses and are therefore expected to have higher f(RH) based on Fig. 4. The WP air masses have coarser particles, based on their SAE values (Fig. 3a), and have a marine origin (with higher DMA mass fraction, Fig. 3b) with high sea salt fraction (54%, Fig. 3b), contributing to it having the highest median f(RH) amongst the air masses."*

(7) Table 1 lists the measurements of cloud particles but there was no discussion of CCN or cloud particles in the manuscript. I recommend deleting the final two rows in the table, listing the cloud particle measurements

*Response: We included the FCDP and 2D-S in the data table because they were used to filter for cloud-free conditions.*

*The text in section 2.2.1 states this. "Cloud-free conditions were identified to ensure the highest quality aerosol data using a cloud flag product based on measurements from the fast cloud droplet probe (FCDP) and two-dimensional stereo probe (2DS)."*

*The text in the table (this is now Table 2) was also updated to note that the parameters were used for cloud screening. The following text was added "(data used for cloud screening)" below Cloud particles in the leftmost column of the last two rows of the table (now Table 2, as a table was added to the introduction).*

---

## Author Response (AR2)

*EDITOR*

*Response: We thank the editor for the comments that have improved our manuscript in terms of its accessibility, clarity, and connection to past studies. We provide our responses below in italicized bold font. All changes to the manuscript can be identified in the version submitted using Track Changes.*

Review of Revised Submission "Measurement Report: Characterization of Aerosol Hygroscopicity over Southeast Asia during the NASA CAMP²Ex Campaign" by Lorenzo et al.

Editor

March 1, 2025

Thanks for your revised manuscript. I had another read and found some more minor issues which should be addressed before final publications. See comments below.

- A few clarifications concerning Table 3:

\* In Burgos et al. (2019), different set-ups of f(RH) measurements were summarized. The DOE system provides PM1/PM10 measurements of f(RH) while other systems provide whole-air values of f(RH). For example, in Zieger et al. (2010) for the Arctic (see https://acp.copernicus.org/articles/10/3875/2010/acp-10-3875-2010.html), it is not "<1000nm" (as stated in the column description) but "whole-air". Please revise.

*Response: The table was deleted and is now included in Section 3.1.1 instead. The following text was added to the 4th paragraph of the introduction. "Observed and simulated aerosol hygroscopicity using aforementioned parameters are greater in clean marine air masses ($f(RH=85 \%/RH=40 \%) = 2.10 - 2.30$ for $PM_1$) compared to air masses over land and near terrestrial biogenic sources ($f(RH=85 \%/RH=40 \%) = 1.29 - 2.10$ for $PM_1$), which are secondary organics precursors, and from biomass burning ($f(RH=85 \%/RH=40 \%) = 1.02 – 2.10$ for $PM_1$) (Swietlicki et al., 2008; Duplissy et al., 2011; Petters and Kreidenweis, 2007; Burgos et al., 2019; Titos et al., 2021; Gomez et al., 2018). Only the Arctic has aerosol hygroscopicity values ($f(RH=85 \%/RH=40 \%) = 2.70$ for $PM_1$ and $f(RH=85 \%/RH=40 \%) = 3.00$ for whole-air samples) greater than in clean marine regions (Delene and Ogren, 2002; Zieger et al., 2010)."*

\* Why do you add values for "new particle formation"? I don't really see this discussed?

*Response: The row for new particle formation has been removed from Table 1.*

\* As also the first reviewer pointed out: It would make more sense to have a more specific table with f(RH) and g(RH) values for marine and marine polluted (or biomass burning) environments. Please also use original references. You could add your observations to the same table for a better comparison ("this study").

*Response: A revised table that includes the CAMP²Ex air mass f(RH) values was added in Section 3.1.1. and is now named Table 2. The table with instrument specifications is now labeled as Table 1. The original references for the f(RH) from Past Studies are now included in the table caption.*

*Table 2: Hygroscopicity values of various aerosol types based on selected summary studies for measured g(RH) (Swietlicki et al., 2008) and f(RH) (Eldering et al., 2002; Doherty et al., 2005; Liu and Li, 2018; Dumka et al., 2017; Gogoi et al., 2015; Burgos et al., 2019; Titos et al., 2021) and modeled $\kappa$ (Pringle et al., 2010) and CAMP²Ex f(RH) values for air masses (Hilario et al., 2021): Maritime Continent (MC), East Asia (EA), peninsular Southeast Asia (PSEA), West Pacific (WP), and Other, and their associated source.*

| *Aerosol Type* | *Past Studies* | | | *Air Masses Affecting CAMP²Ex* |
| | *g(RH=90%)\** *(10 - 500 nm)* | *f(RH=85%/RH=40%)\*\** *(< 1000 nm)* | *$\kappa$\*\*\*\** *(5 - 500 nm)* | *f(RH=82 ± 10 %/RH<40%)\*\** *(< 5000 nm)* |
| --- | --- | --- | --- | --- |
| *Clean Marine* | *1.62 - 2.14* | *2.10 - 2.30* | *0.92* | *1.49 (WP)* |
| *Polluted Marine* | *1.00 - 1.76* | *1.73 - 2.20* | *0.59* | *1.23 (Other)* |
| *Urban Continental* | *1.00 - 1.68* | *1.38 - 1.60* | *0.21 - 0.36* | *1.20 - 1.38 (EA, PSEA)* |
| *Aged Biomass* | *1.15 - 1.30* | *1.02 - 2.1\*\*\** | *< 0.1* | *1.05 (MC)* |

*\*Mean values from various ground-based sites*

*\*\*Median values*

*\*\*\*Gomez et al., 2018 mean (f(RH=85%/RH=25%))*

*\*\*\*\*Mean surface values*

- Line 173-176: There is some repetition here about the verification using hydrophilic PSLs. Please combine. In addition, are they really hydrophilic? It is a plastic and should not be hygroscopic. Please double-check what you used.

*Response: The sentences have been combined "System response was verified in flight by introducing hydrophobic PSL spheres into the sample stream to ensure that the nephelometers were consistent and an f(RH) value of 1.0 is observed." And the sentence with the hydrophilic PSLs was deleted.*

- In your equations, you are using for the scattering enhancement an italic "f". Please also use it within the text, to make sure that the scattering enhancement is a function.

*Response: The f in f(RH) has been replaced by an italicized "f" for the 139 instances that f(RH) appears in the text and the captions.*

- For all figures showing boxplots/errorbars and linear regressions: Please add information on how they were retrieved and what is shown.

*Response: This was added to the caption for Figures 2, 3, and 6 "(boxes correspond to the 25th to 75th percentile of the data, the red line is the 50th percentile, the whiskers represent 1.5 \* the interquartile range, the red crosses are the outliers, and when the notches in the boxplots do not overlap there is 95 % confidence that the medians of the boxplots are different)".*

- Some of the figure captions could be improved by using a descriptive title (and not start with the temporal resolution of a certain quantity). See https://www.atmospheric-chemistry-and-physics.net/submission.html#figurestables

*Response: The following captions were edited:*

*Figure 2. "Distribution of f(RH=82 % ± 10 %/RH<40 %) 1 s data at 550 nm"*

*Figure 3. The following descriptive title was added "Physicochemical properties of air masses." The words "1 s" were removed from the first sentence of the caption.*

*Figure 4. The following descriptive title was added "Effect of organic mass fraction on f(RH)."*

*Figure 5. The following descriptive title was added "Aerosol types based on particle optical data."*

*Figure 6. The following descriptive title was added "Physicochemical properties of air masses with sub-1 f(RH)."*

*Figure 8. The following descriptive title was added "Effective particle density and f(RH)."*

*Figure 10. The following descriptive title was added "Volume size distributions for the tropical cyclone-induced convection case study." And the words "the tropical cyclone-induced convection case study" were removed from the next sentence.*

*Figure 11. The following descriptive title was added "Volume size distributions for the shallow convection case study." And the words "the shallow convection case study" were removed from the next sentence.*

- In Figure 1, please add the corresponding wavelength of f(RH) to the caption. The counts relate to what time average? I am also a bit confused by the RH histogram and the f(RH) values in the

histogram above: Is f(RH) shown at variable RH (as shown in the RH histogram below) or shown at f(RH=82/40) as mentioned in the text. Please make this clear in the figure and caption.

*Response: The wavelength "at 550 nm" was added after the two instances of f(RH) mentioned in the caption. The text in Figure 1 was updated to note the RH values used to derive f(RH) f(RH=82 % ± 10 %/RH<40 %). The word "ambient" before "relative humidity (RH" ) was also added for a clearer description of the plot of ambient RH measurements that correspond to the f(RH) data. The captions in Figures 2, 4 to 9 and 13 were also updated to include "RH=82 % ± 10 %/RH<40 %"*

- Figure 2: Same here, mention which RH and wavelength is shown here.

*Response: The RH and wavelength values for f(RH) were added accordingly to this figure caption as well for the figure captions in Figures 4 to 9 and 13.*

- Figure 3: Please describe properly panel b (e.g. acronyms, y-axis, etc).

*Response: The caption was updated and now includes the instruments ("aerosol mass spectrometer (AMS)", "soot photometer (SP2)", and "particle-into-liquid sampler (PILS) and ion chromatograph") used to derive the aerosol mass fractions. The air mass names and their acronyms, which are listed on the y-axis, were also explicitly stated in the updated the caption (": Maritime Continent (MC), East Asia (EA), Peninsular Southeast Asia (PSEA), West Pacific (WP), and Other").*

- Figure 4: Please add the used regression type (orthogonal?) to the caption. As a side note: This is a nice result, you could compare your findings to previous observed f(RH)-organic relationships e.g. given by Quinn et al. (2005), Zhang et al. (2015), and Zieger et al. (2015). See also Figure 3 in Burgos et al. (2020) where these kind of findings are later used to evaluate models.

*Response: The caption was updated: Scatter plots of 30 s averaged f(RH=82 % ± 10 %/RH<40 %) and corresponding organic mass fraction contribution to total submicrometer aerosol mass (sum of organic, $SO_4^{2-}$, $NO_3^-$, $NH_4^+$, $Cl^-$, and black carbon (BC) mass concentration) "and (red) linear regression line and (upper right) corresponding equation and goodness of fit"*

*The following text was added to Section 3.1.1:*

*"The relationship between f(RH) and organic mass during CAMP²Ex is similar to those in the Indian ocean during INDOEX (based on calculated f(RH) from estimated γ) (Quinn et al., 2005), which is similarly affected by the outflow of biomass burning."*

*"This slope for the air masses coming from EA is less steep than the f (RH=85 %/RH=40 %) at 550 nm to organic mass fraction relationship (-1.20 ± 0.04) from a ground station in the populated and growing region of the Yangtze River Delta in China (Zhang et al., 2015), where*

*organic contributions are lower and mean f(RH) values are higher (1.58 ± 0.12) than the median f(RH) (1.38) during CAMP2Ex for air massed coming from EA."*

*The third paragraph in the conclusion was updated to include findings from the sources mentioned above. "Without organics, the baseline f(RH) (1.38 – 1.55, Fig. 4), is still lower than measurements from other areas (i.e. ~1.8 from the Yangtze River Delta in China and from INDOEX over the Indian Ocean, 2.19 from SEAC$^4$RS in and around the U.S., ~2.6 from ICARTT over North America and Europe, ~3 from ACE-ASIA over the Yellow Sea and Sea of Japan, and ~3.5 from Melpitz, Germany) (Zhang et al., 2015; Quinn et al., 2005; Shingler et al., 2016a; Zieger et al., 2014) probably due to the presence of elemental carbon, which is the second most dominant aerosol type during the campaign. The unique relationship between organic matter and hygroscopicity in the CAMP$^2$Ex region is important information that can help in model evaluations (Burgos et al., 2020)."*

- Figure 5: Please make sure that the corresponding wavelengths are mentioned in the caption.

*Response: The wavelengths were added accordingly: (AAE, 470-660 nm) and (SAE, 450-700 nm).*

- Figure 8: Regression type?

*Response: The following was added to the caption "and with (red) linear regression line and (upper left) corresponding equation and goodness of fit"*

- In the data availability statement, could you please be more detailed where the individual datasets can be found? For example, I could not find the f(RH) data on the website stated by just looking at the file names.

*Response: The data availability statement has been updated to include more details. "CAMP$^2$Ex data archived per research flight and parameter type can be found at: https://doi.org/10.5067/Airborne/CAMP2Ex_Aerosol_AircraftInSitu_P3_Data_1 (P-3 In-Situ Aerosol Data), https://doi.org/10.5067/Airborne/CAMP2Ex_Cloud_AircraftInSitu_P3_Data_1 (P-3 In-Situ Cloud Data), https://doi.org/10.5067/Airborne/CAMP2Ex_MetNav_AircraftInSitu_P3_Data_1 (P-3 In-Situ Meteorological and Navigational Data), and https://doi.org/10.5067/Airborne/CAMP2Ex_TraceGas_AircraftInSitu_P3_Data_1 (P-3 In-Situ Trace Gas Data)."*

References:

Burgos, M. A., Andrews, E., Titos, G., Benedetti, A., Bian, H., Buchard, V., Curci, G., Kipling, Z., Kirkevåg, A., Kokkola, H., Laakso, A., Letertre-Danczak, J., Lund, M. T., Matsui, H., Myhre, G., Randles, C., Schulz, M., van Noije, T., Zhang, K., Alados-Arboledas, L., Baltensperger, U.,

Jefferson, A., Sherman, J., Sun, J., Weingartner, E., and Zieger, P.: A global model–measurement evaluation of particle light scattering coefficients at elevated relative humidity, Atmos. Chem. Phys., 20, 10231–10258, https://doi.org/10.5194/acp-20-10231-2020, 2020.

Quinn, P. K., Bates, T. S., Baynard, T., Clarke, A. D., Onasch, T. B., Wang, W., Rood, M. J., Andrews, E., Allan, J., Carrico, C. M., Coffman, D., and Worsnop, D.: Impact of particulate organic matter on the relative humidity dependence of light scattering: A simplified parameterization, Geophys. Res. Lett., 32, L22809, https://doi.org/10.1029/2005GL024322, 2005.

Zhang, L., Sun, J. Y., Shen, X. J., Zhang, Y. M., Che, H., Ma, Q. L., Zhang, Y. W., Zhang, X. Y., and Ogren, J. A.: Observations of relative humidity effects on aerosol light scattering in the Yangtze River Delta of China, Atmos. Chem. Phys., 15, 8439–8454, https://doi.org/10.5194/acp-15-8439-2015, 2015.

Zieger, P., Fierz-Schmidhauser, R., Poulain, L., Müller, T., Birmili, W., Spindler, G., Wiedensohler, A., Baltensperger, U., and Weingartner, E.: Influence of water uptake on the aerosol particle light scattering coefficients of the Central European aerosol, Tellus B, 66, 22716, https://doi.org/10.3402/tellusb.v66.22716, 2014.

Zieger, P., Aalto, P. P., Aaltonen, V., Äijälä, M., Backman, J., Hong, J., Komppula, M., Krejci, R., Laborde, M., Lampilahti, J., de Leeuw, G., Pfüller, A., Rosati, B., Tesche, M., Tunved, P., Väänänen, R., and Petäjä, T.: Low hygroscopic scattering enhancement of boreal aerosol and the implications for a columnar optical closure study, Atmos. Chem. Phys., 15, 7247–7267, https://doi.org/10.5194/acp-15-7247-2015, 2015.